# Functional cell types in the mouse superior colliculus

Ya-tang Li[1,2]*, Markus Meister[1,3]*

[1]Division of Biology and Biological Engineering, California Institute of Technology, Pasadena, CA, United States; [2]Chinese Institute for Brain Research, Beijing, China; [3]Tianqiao and Chrissy Chen Institute for Neuroscience, California Institute of Technology, Pasadena, CA, United States

**Abstract** The superior colliculus (SC) represents a major visual processing station in the mammalian brain that receives input from many types of retinal ganglion cells (RGCs). How many parallel channels exist in the SC, and what information does each encode? Here, we recorded from mouse superficial SC neurons under a battery of visual stimuli including those used for classification of RGCs. An unsupervised clustering algorithm identified 24 functional types based on their visual responses. They fall into two groups: one that responds similarly to RGCs and another with more diverse and specialized stimulus selectivity. The second group is dominant at greater depths, consistent with a vertical progression of signal processing in the SC. Cells of the same functional type tend to cluster near each other in anatomical space. Compared to the retina, the visual representation in the SC has lower dimensionality, consistent with a sifting process along the visual pathway.

## Editor's evaluation

This important paper will be of interest to neuroscientists interested in how the representation of sensory information is refined between the sensory periphery and more central areas. The work provides compelling evidence for a much greater diversity of functional cell types in the superior colliculus than previously suggested, and that the functional organization of cell types in the superior colliculus is distinct from that of the retina.

*For correspondence:
yatangli8@gmail.com (Y-tL);
meister4@mac.com (MM)

## Introduction

Parallel processing of information by the brain operates at three different levels: single neurons, cell types, and neural pathways. Processing of visual information at the cell-types level starts at the retina, where the signal is split into ~15 types of bipolar cells (*Shekhar et al., 2016*), ~63 types of amacrine cells (*Yan et al., 2020*), and ~40 types of retinal ganglion cells (RGCs; *Roska and Meister, 2014*; *Sanes and Masland, 2015*; *Baden et al., 2016*). The RGCs send the processed information directly to the superior colliculus (SC), an evolutionarily conserved structure found in all vertebrates (*Isa et al., 2021*; *Basso and May, 2017*). The SC serves as an important visual center and also plays a vital role in coordinating animal behavior (*Wheatcroft et al., 2022*).

In the rodent, ~90% of RGCs project to the superficial layer of the SC (*Ellis et al., 2016*), and each SC neuron receives inputs from about six RGCs (*Chandrasekaran et al., 2007*). It remains unclear how the information is transformed at this stage and how many cell types exist in the SC. As in other brain areas, a solid classification of cell types in this circuit would support a systematic study of its function (*Zeng and Sanes, 2017*). By classic criteria of cell morphology and physiology, prior work has distinguished about five cell types in the retino-recipient superficial layer (*Langer and Lund, 1974*; *May, 2006*; *Gale and Murphy, 2014*; *Wang et al., 2010*; *De Franceschi and Solomon, 2018*). Differential

expression of molecular markers has been used to describe about ten types in the superficial SC (*Byun et al., 2016*). By contrast, recent work on the primary visual cortex (V1) identified 46 types of neurons based on morphology and electrophysiology (*Gouwens et al., 2019*). Because the major inputs to the superficial SC are from the retina and V1, we hypothesize that the number of functionally distinct cell types in the SC has been underestimated.

The present work aims to identify functional cell types in the superior colliculus by virtue of their responses to a large set of visual stimuli. These include a panel of stimuli that successfully separated ~40 types of retinal ganglion cells, confirming many classes previously known from anatomical and molecular criteria (*Baden et al., 2016*). By two-photon calcium imaging, we recorded neuronal responses from the posterior-medial SC of behaving mice while leaving the cortex intact. We classified cell types based on their response to the high-dimensional visual stimulus using unsupervised learning algorithms. We included several transgenic mouse strains that label subsets of SC neurons based on gene expression patterns. The evidence points to ~24 functional types that come in two major classes: one closely related to retinal responses, the other distinct. We report on the anatomical organization of these functional types, their relation to molecular cell types, and their progression throughout layers of the superficial SC. By comparing the space of visual features encoded in the SC to that in the retina, one finds that the superior colliculus already discards substantial information from the retinal output.

## Results

### Single-cell imaging reveals diverse neuronal responses to a set of visual stimuli

To investigate the functional diversity of SC neurons, we imaged neuronal calcium responses to a battery of visual stimuli using two-photon microscopy in head-fixed awake mice (*Figure 1A*). To maintain the integrity of the overlying cortex, one is limited to the posterior-medial SC that corresponds to the upper lateral visual field (*Feinberg and Meister, 2015*; *Figure 1B and C*). We recorded more than 5000 neurons from 41 image planes in 16 animals from different genetic lines, including wild-type, Vglut2-Cre, Vgat-Cre, Tac1-Cre, Rorb-Cre, and Ntsr1-Cre mice. In the Cre lines, the calcium indicator was restricted to the neurons expressing the respective transgene.

We presented a battery of visual stimuli (*Figure 1D*) chosen to probe spatio-temporal integration, color-sensitivity, and movement processing (see Visual stimulation in Materials and methods for detail). Included was a 'chirp' stimulus that modulates the intensity on the cell's receptive-field over both frequency and amplitude; the full-screen 'chirp' was previously employed in the classification of retinal ganglion cells (*Baden et al., 2016*). Neurons in any given image plane showed robust and diverse responses to these stimuli (*Figure 1C–D*). The animals were positioned on a circular treadmill but remained stationary during most of the visual stimulation. Because locomotion barely modulates the visual responses of SC neurons (*Savier et al., 2019*), we did not consider further the effects of movements.

### Superficial superior colliculus comprises at least 24 functional cell types

To classify cell types based on their functional properties, we first performed a sparse principal component analysis on the raw response traces (*Figure 2—figure supplement 1A*; *Mairal et al., 2009*), which led to a 50-dimensional feature vector for each neuron. Then we added 4 designed features that describe different aspects of the response: a habituation index (HI) computed from repeated stimuli, a direction selectivity index (DSI), an orientation selectivity index (OSI), and a motion selectivity index (MSI, see Materials and methods). We focused on 3414 neurons that responded reliably to visual stimuli (signal-to-noise ratio >0.35, see definition in Materials and methods *Equation 2*), and searched for clusters in the 54-dimensional feature space by fitting the data (3414 cells × 54 features) with a Gaussian mixture model, varying the number of clusters in the mixture (*Figure 2*).

The quality of each clustering was assessed with the Bayesian information criterion (BIC), which addresses concerns about overfitting by balancing the goodness of fit with generalizability. By this measure, the distribution of cells in feature space was best described with a Gaussian mixture of 24 components (*Figure 2B*), suggesting there are 24 functional types of neurons in the surveyed population. Because the optimum in the BIC curve was rather broad (*Figure 2B*), and one could make a

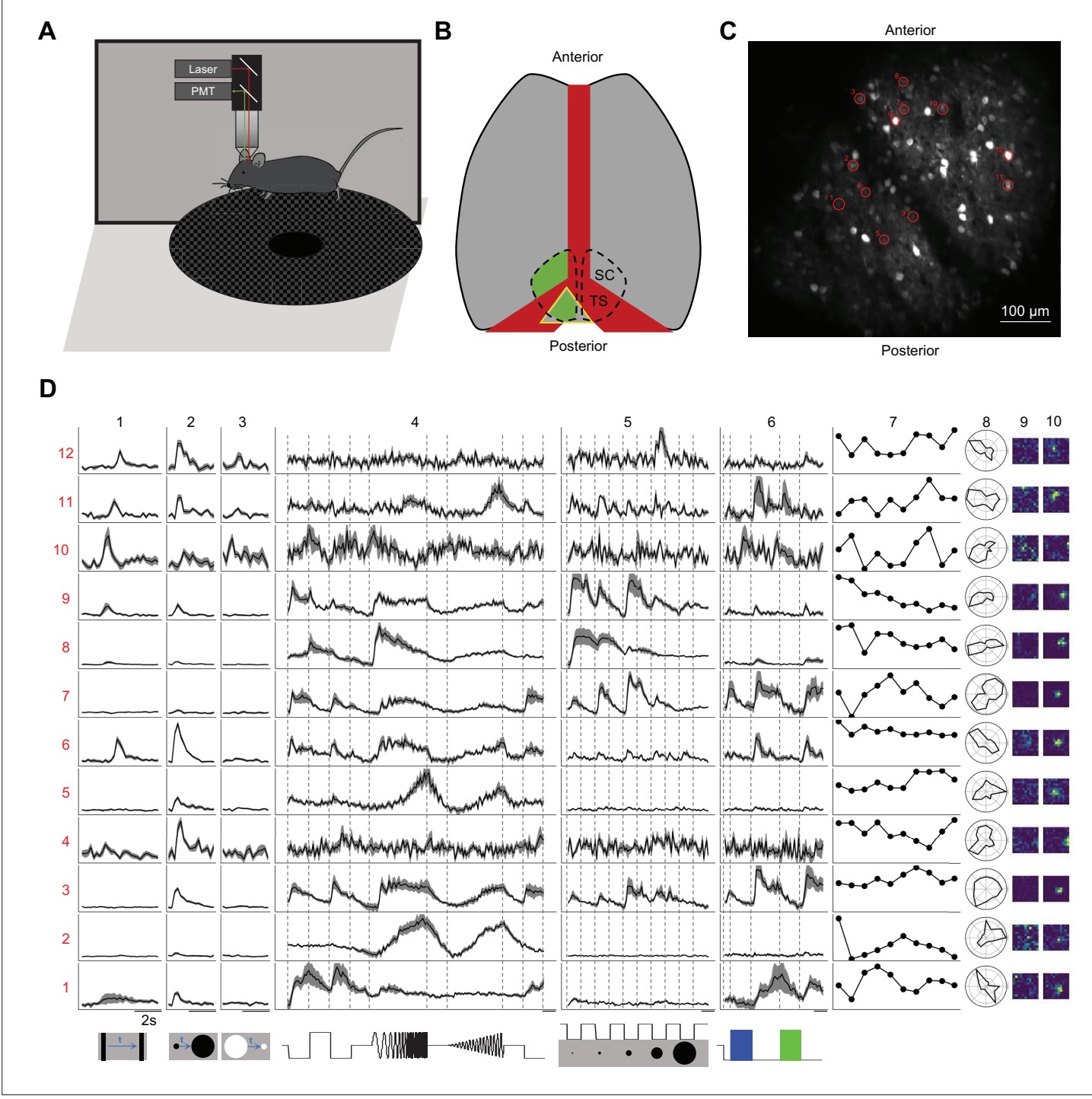

**Figure 1.** Two-photon imaging reveals diverse visual responses in awake mouse superior colliculus. (**A**) Schematic of the experimental setup. Mice were head-fixed and free to run on a circular treadmill. Visual stimuli were presented on a screen. Neuronal calcium activity was imaged using two-photon microscopy. PMT, photomultiplier tube. (**B**) Schematic of mouse brain anatomy after insertion of a triangular transparent plug to reveal the posterior-medial part of the superior colliculus underneath the transverse sinus. TS: transverse sinus. (**C**) A standard deviation projection of calcium responses to visual stimuli in one field of view. (**D**) Response profiles of 12 neurons (rows) marked in C to visual stimuli in the bottom row. Columns 1–6 are time-varying calcium responses to a moving bar, expanding and contracting disks, a 'chirp' stimulus with modulation of amplitude and frequency, spots of varying size, and blue and green flashes. Gray shading indicates the standard error across identical trials. Each row is scaled to the maximal response. Scale bars: 2 s. Subsequent columns show processed results: (7) response amplitude to an expanding black disc on 10 consecutive trials. (8) polar graph of response amplitude to moving bar in 12 directions. (9 and 10) Receptive field profiles mapped with small squares flashing On or Off.

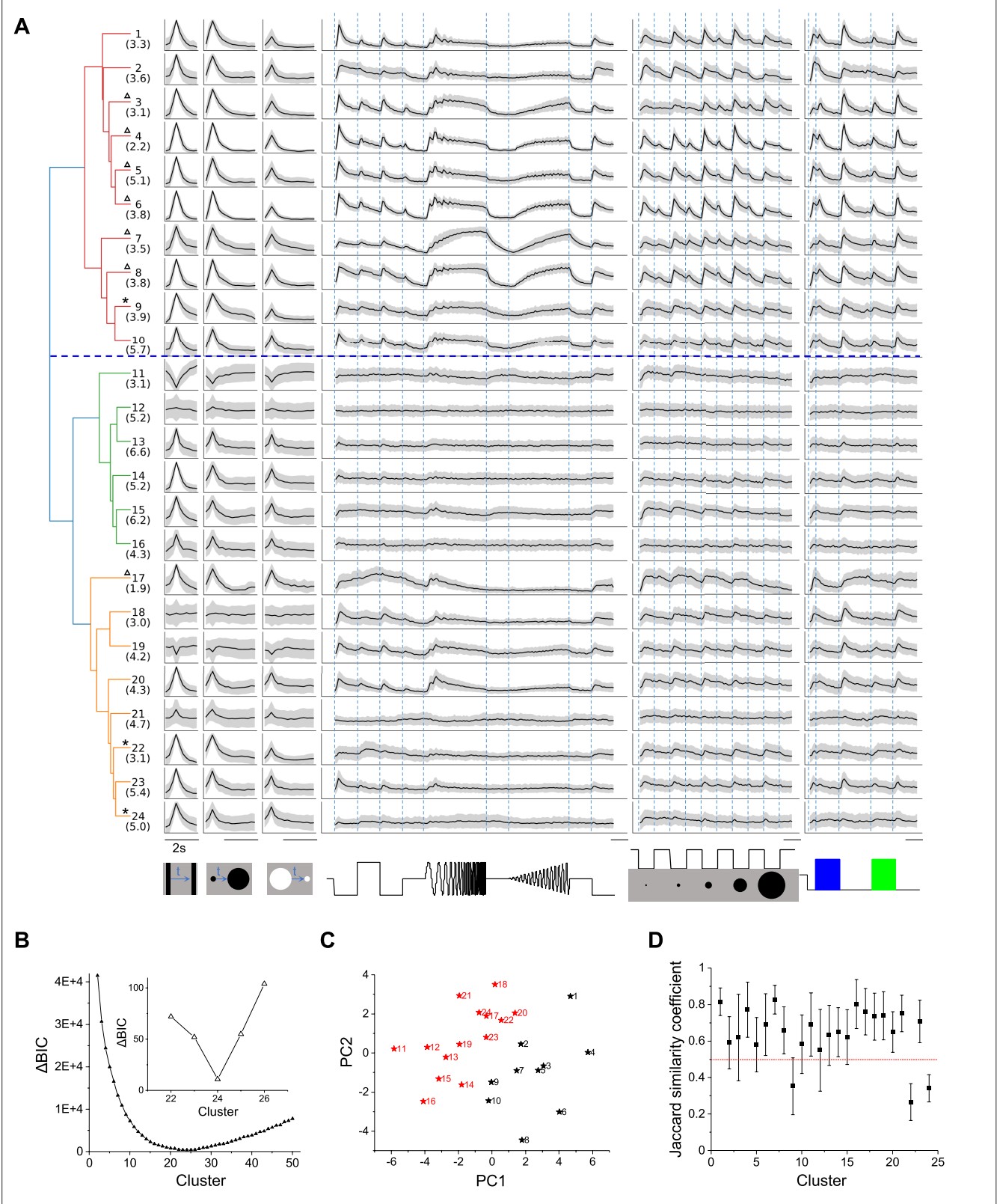

**Figure 2.** Twenty-four functional cell types in the mouse SC. (**A**) Dendrogram of 24 clusters based on their distance in feature space. For each type, this shows the average time course of the neural response to the stimulus panel. Grey: standard deviation. The vertical scale is identical for all types and stimulus conditions. Blue dashed line separates groups 1 and 2. Numbers in parentheses indicate percentage of each type in the dataset. Stars mark the unstable clusters with JSC< 0.5 (see panel D). Triangles mark the clusters where more than half neurons are contributed by one mouse. Scale bars:

*Figure 2 continued on next page*

*Figure 2 continued*

2 s. (**B**) Relative Bayesian information criterion (ΔBIC) for Gaussian mixture models with different numbers of clusters. (**C**) The center of each cluster in the first two principal axes of feature space. Black and red colors label Groups 1 and 2, respectively. (**D**) Jaccard similarity coefficient (JSC) between the full dataset and subsets (Mean ± SD).

The online version of this article includes the following figure supplement(s) for figure 2:

**Figure supplement 1.** Clustering and validating (related to *Figure 2*).

**Figure supplement 2.** Dendrogram of 24 clusters showing normalized temporal profiles (related to *Figure 2*).

**Figure supplement 3.** Example responses of single neurons in each type to visual stimuli (related to *Figure 2*).

case for both fewer or more clusters, we followed up by testing the stability of each cluster: We fitted various sub-samples of the data set and assessed how well the resulting clusters correspond to those in the full set, using a number of established statistics (see Materials and methods and *Figure 2— figure supplement 1*). It emerged that 3 of the 24 clusters are somewhat unstable (*Figure 2D*); they are marked as such in *Figure 2A*. Overall the stability of the cluster definitions matched or exceeded those in related studies of neuronal cell types (*Gouwens et al., 2019*; *Baden et al., 2016*). For the purpose of subsequent analysis we will adopt this division into 24 types as suggested by the BIC.

The hierarchical relationship between these functional types is illustrated by the dendrogram in *Figure 2A*, which is based on the distances in feature space between cluster centers. The first branching of the dendrogram splits the types into two groups (*Figure 2A, C*, *Figure 2—figure supplement 2* and *Figure 2—figure supplement 3*), which we will call Group 1 (types 1–10) and Group 2 (types 11–24). Group 1 further splits into Group 1a (types 1–6) and 1b (types 7–10).

Group 1 (types 1–10) is distinct from Group 2 (11-24) in that it responds more strongly to the chirp stimuli (flashes and sinusoidal modulations in *Figure 2A*, 0.14 ± 0.05 vs. 0.04 ± 0.04, $p<0.001$, two-sample $t$-test). All types in Group 1 prefer the expanding black disc over the receding white disc. Almost all these types are excited by both On and Off stimuli in the chirp in the receptive field. Also they prefer large spots to small spots, type 2 being a notable exception. Within Group 1, types 1–6 (Group 1a) are distinct from types 7–10 (Group 1b) in their response to sinusoid flicker: Group 1a prefers the low frequencies, whereas Group 1b responds over a wider range ($p=0.02$, Wilcoxon rank-sum test). Type 7 in particular rejects the low flicker frequencies.

In Group 2, the response to chirp stimuli is generally weak compared to the moving stimuli. These types respond well to small spots, unlike the Group 1 types. Other response properties in Group 2 are more diverse, and some of these types have been noted previously. For example, Types 11 and 19 stand out in that moving stimuli suppress their activity (*Ito et al., 2017*). Several types (11, 15, 17) are suppressed by the sinusoid flicker (*Ito et al., 2017*); 11 and 15 also show rebound after cessation of that stimulus. Type 14 responds strongly to moving stimuli but hardly at all to the entire chirp. Type 18 is remarkably insensitive to any moving stimulus.

What are the distinguishing features in their visual responses? *Figure 3B* distills the responses to the stimulus palette into 15 indices (see Materials and methods) that help to characterize each type (only values significantly different from zero are shown, see *Figure 3—figure supplement 1A* for all values). For some indices, *Figure 3A* shows violin or bar plots for each type. As a rule, almost all the types are sensitive to moving stimuli, like the traveling dark bar and the expanding black disc (RtM in *Figure 3B*). For many types, these were the stimuli that elicited the strongest response.

Some of these features of the visual response were highly correlated with each other ($p<0.05$), in that they co-varied in the same or opposite directions across types (*Figure 3C*, *Figure 3—figure supplement 1B*). For example, direction selectivity (DSI) and orientation selectivity (OSI) tend to be strong in the same cell type (types 9, 14, 20, 24). Another strong correlation exists between the preferred stimulus size (BSS) and the response during recovery from the frequency and amplitude chirps (RaFM and RaAM).

## Neurons of the same type cluster in anatomical space

In the retina, ganglion cells come in ~40 different types (*Sanes and Masland, 2015*), and they tile the surface in a so-called mosaic arrangement. Neurons of the same type are spaced at regular distances from each other. Neurons of different types are distributed more or less independently (*Roy et al., 2021*); therefore a ganglion cell's nearest neighbor is almost always of a different type. The presumed

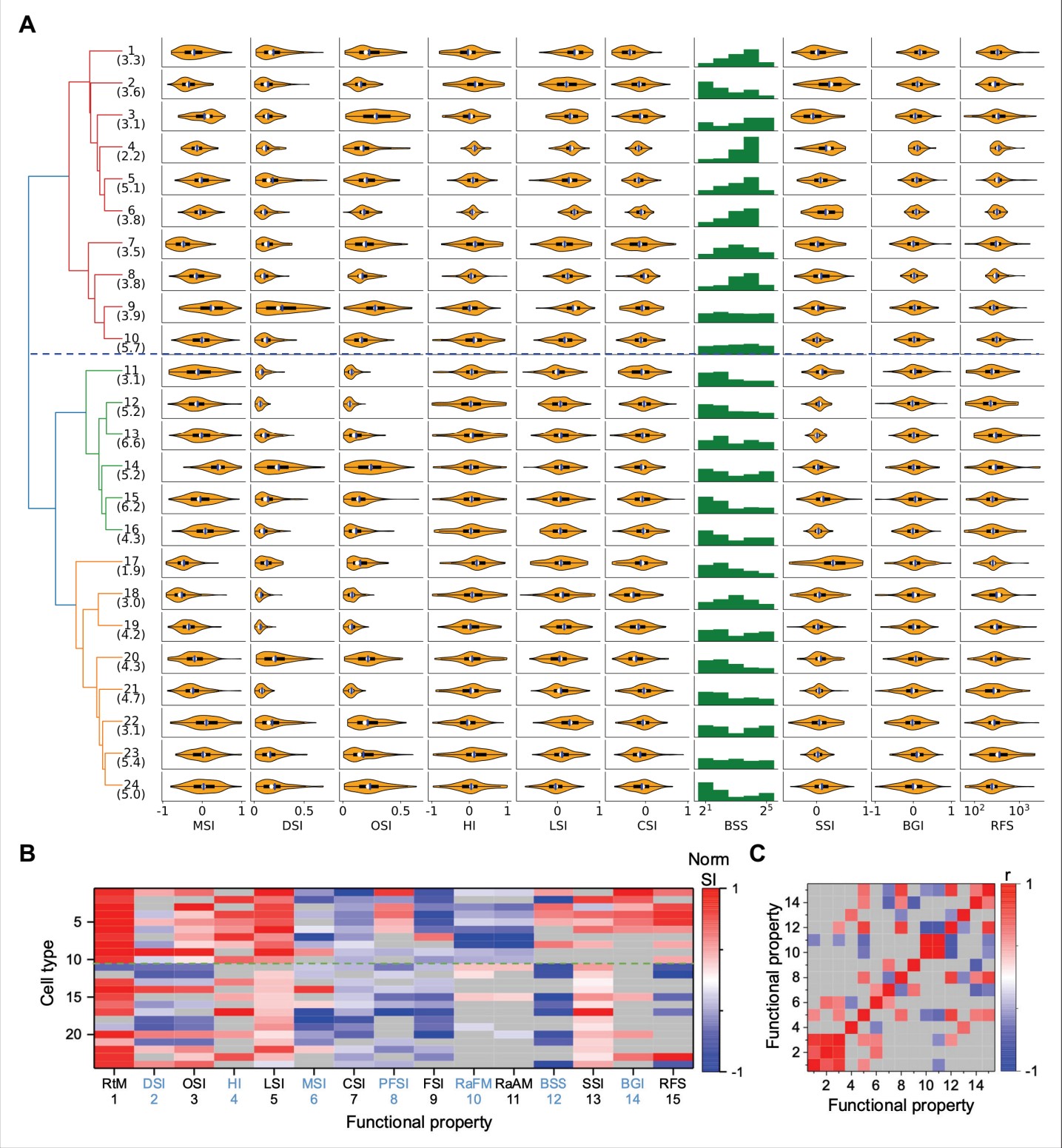

**Figure 3.** Functional diversity among different types. (**A**) Violin plots or histograms of various response indices: motion selectivity index (MSI), direction selectivity index (DSI), orientation selectivity index (OSI), habituation index (HI), looming selectivity index (LSI), contrast selectivity index (CSI), best stimulus size (BSS), surround suppression index (SSI), blue green index (BGI), and receptive field size (RFS). Blue dashed line separates Group 1 and Group 2. (**B**) Normalized selectivity index (SI, normalized for each column) of functional properties represented by different cell types (See Materials and methods). RtM: response to motion; PFSI: peak-final selectivity index; FSI: frequency selectivity index; RaFM: response after frequency modulation; RaAM: response after amplitude modulation. Gray: values that are not significantly different from 0 ($p \geq 0.05$, one-sample $t$-test). Green dashed line

*Figure 3 continued on next page*

*Figure 3 continued*

separates Group 1 and Group 2. (**C**) Pearson's correlation coefficients of the representation between pairs of functional properties. Gray: non-significant correlations ($p \geq 0.05$).

The online version of this article includes the following figure supplement(s) for figure 3:

**Figure supplement 1.** Functional properties of different cell types (related to *Figure 3*).

purpose of this arrangement is to ensure uniform coverage such that every location in the visual field has access to each of the types of retinal ganglion cell. Because the retina projects directly to the SC, we investigated whether neurons there are also organized for uniform coverage.

*Figure 4A* illustrates SC neurons in a single image plane, labeled according to functional type (for more examples see also *Figure 4—figure supplement 1A*). Several features are immediately apparent. First, cells of a given type do not repel each other; in fact, the nearest neighbor is often a neuron of the same type. Second, the types do not cover space uniformly. Some types are segregated from each other (e.g. 7 and 21) whereas others overlap in space (e.g. 14 and 24).

To pursue these spatial arrangements in greater detail, we computed for each functional type the spatial autocorrelation function (also called 'density recovery profile'): This is the average density of neurons plotted as a function of distance from another neuron of the same type (*Zhang et al., 2012*; *Rodieck, 1991*). When applied to retinal ganglion cells, this function shows a pronounced hole of zero density at short distances. Here, instead, the density remains high down to a distance of 10 μm, which is the typical diameter of a soma (*Figure 4B*). In fact for several of these SC types the density is highest just 1–2 cell diameters away.

Over larger distances, the autocorrelation function decays gradually (*Figure 4C*), whereas one would expect a flat curve if the cells appeared at uniform density. The density drops to half the peak value at a radius of 150–250 μm. This suggests that neurons of a given type form patches of 300–500 μm diameter. Note this accords with a similar patchy organization found previously for certain functional parameters, like the preferred orientation (*Feinberg and Meister, 2015*) or preferred direction of motion (*de Malmazet et al., 2018*; *Li et al., 2020*).

We considered a potential source of error that could give the mistaken appearance of a patchy organization: A functional type might fortuitously be limited to just one recording session, owing to some peculiarity of that animal subject, and so would appear only in the visual field covered during that session. This is not the case: *Figure 4—figure supplement 2* shows that recordings from different mice confirm the same functional type. Furthermore, a single recording session reveals separate patches of different types (*Figure 4A*, *Figure 4—figure supplement 1A*).

An important scale by which to judge this spatial organization is the size of the receptive field (RF). In the retina, the mosaic arrangement spaces the ganglion cells about one RF diameter apart, so that the RFs of neurons from the same type have little overlap. In the SC that is clearly not the case. From the autocorrelation functions (*Figure 4B*), we compared the average density of cells within 0.5 RF diameters to the density at 0.5–1.0 RF diameters. In the SC, that ratio is greater than 1 for all functional types and often close to 2 (*Figure 4D*). By comparison, for the W3 type of retinal ganglion cell (*Zhang et al., 2012*) that ratio is only 0.4.

In the retina, the arrangement of one functional type is almost entirely independent of the other types (*Wässle et al., 1981*; but see *Roy et al., 2021*). In the SC we found a strong anticorrelation: Within 50 μm of a given cell, the cells of the same type occurred at greater-than-average density, but cells of the other types at lower density (*Figure 4E*). One might say that each functional type tends to displace the others. To illustrate this further, we calculated the normalized distance between all pairs of cell types (*Figure 4F*). Twenty-one types show significant spatial separation from at least one other cluster (*Figure 4—figure supplement 1B*, $p < 0.05$, bootstrap analysis).

The functional properties of neurons also varied along the depth axis of the superior colliculus (*Figure 4G*). The individual response parameters changed in only subtle ways; for example one can find more neurons with larger receptive fields at greater depth but on average deeper neurons preferred smaller spot stimuli. However, at the level of identified types (which takes many response parameters into account) the differences were more pronounced. In particular, neurons in the upper 100 μm were composed primarily of types 1–10 (Group 1, 56%) while neurons deeper than 100 μm belonged primarily to types 11–24 (Group 2, 68%, $p < 0.001$, chi-square test, *Figure 4H*, *Figure 4—figure supplement 1C–D*).

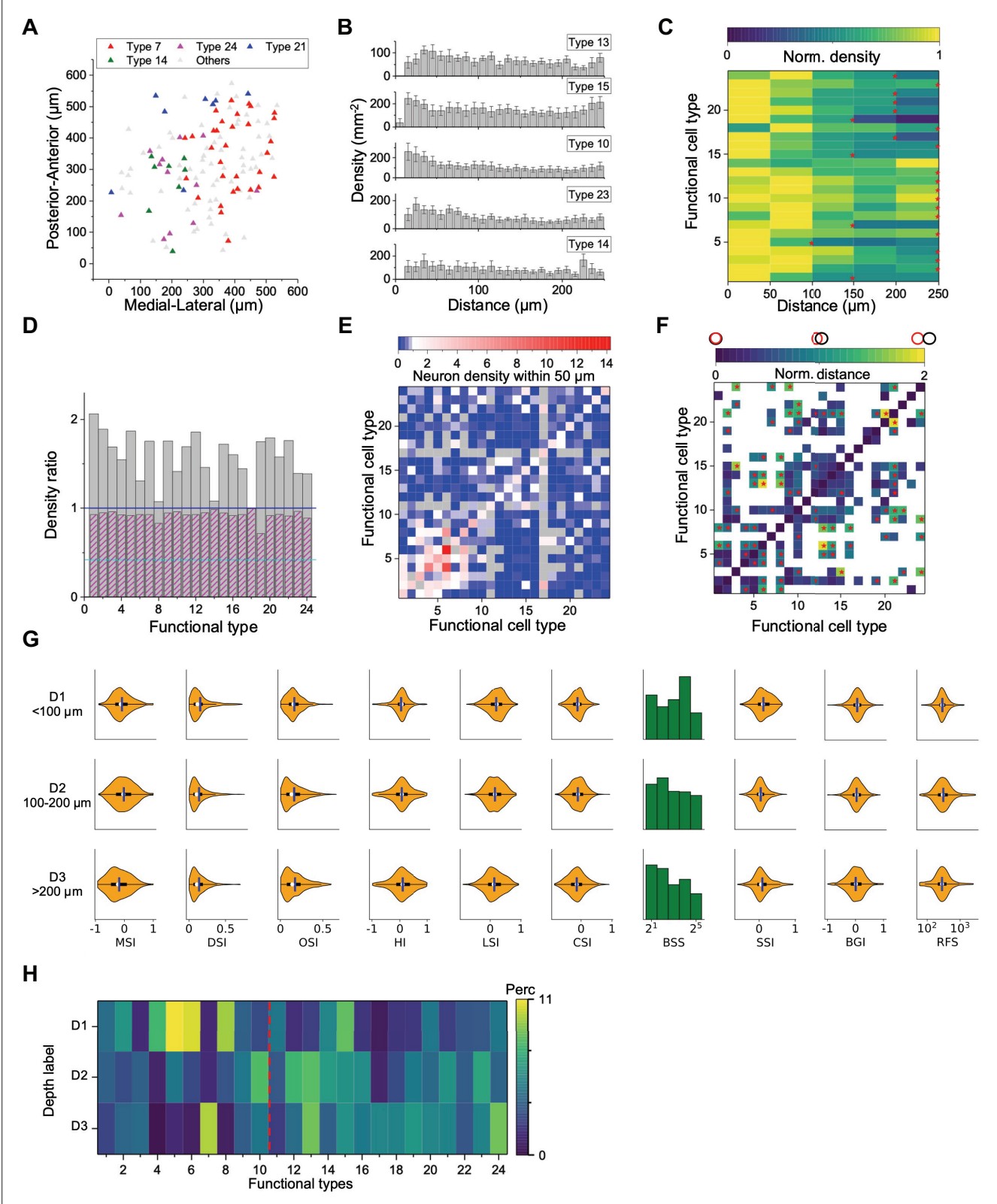

**Figure 4.** Spatial organization of functional cell types. (**A**) Anatomical locations of four types of neurons from one sample recording. (**B**) Averaged density recovery profile (DRP) of five example types for all imaging planes. Error bars denote SEM. (**C**) Density of neurons of a given functional type at various distances from a neuron of the same type. Red stars mark the smallest radius within which the neuron density is larger than half of the peak density. (**D**) Decay of the DRP for each functional type. Gray bars: the ratio between the density at a distance of 0–0.5 RF diameters and the density at

*Figure 4 continued*

0.5–1 RF diameters. Magenta bars: Same but for cells from all the other types. Cyan line: The value for W3 retinal ganglion cells (*Zhang et al., 2012*). (**E**) Density of neurons from different functional types (columns) within 50 μm of a given neuron whose type is indicated by the row. Note the largest density is for cells of the same type except types 8, 11, and 16. Gray: insufficient data to estimate density. (**F**) Normalized anatomical distance between neurons in any two functional types. Red stars indicate significant separation (*p*<0.05, bootstrap analysis). White: insufficient data. (**G**) Functional properties of neurons in three ranges of depth. Display as in *Figure 3A*. (**H**) The percentage of functional types in each depth range. Red dashed line separates Group 1 and Group 2.

The online version of this article includes the following figure supplement(s) for figure 4:

**Figure supplement 1.** Anatomical organization of functional types (related to *Figure 4*).

**Figure supplement 2.** Receptive field center positions for each functional type (related to *Figure 4*).

## Genetically labeled populations comprise multiple functional types

Several experiments focused on superior colliculus neurons with specific molecular identity, taking advantage of existing mouse Cre-lines (*Figure 5*). The Vgat-Cre and Vglut-Cre lines respectively label GABAergic and glutamatergic neurons; Tac1-Cre labels populations of neurons stratified in the superficial SC (*Harris et al., 2014*), Rorb-Cre labels both excitatory and inhibitory neurons in the superficial sublayer (*Byun et al., 2016*; *Gale and Murphy, 2018*); finally Ntsr1-Cre labels a subtype of excitatory neurons that have a wide-field morphology at deeper locations (*Gale and Murphy, 2014*; *Gale and Murphy, 2016*; *Gale and Murphy, 2018*).

Some systematic differences between these genetically labeled populations are apparent already from the chirp responses (*Figure 5A*). For example, excitatory and inhibitory neurons differ at the population level: Vglut + neurons prefer flashed over moving stimuli and spots of small size, whereas Vgat + neurons respond equally to flashed and moving stimuli and prefer large-size spots (*Figure 5B*). Tac1 + neurons have the highest direction selectivity and small receptive fields, whereas Ntsr1 + neurons show strong orientation selectivity. We refrain from analyzing the fine-grained distribution of the 24 functional types in each genetic population, because the experiments did not cover every combination of genetic label and depth and visual field location, leading to possible confounds.

Prior reports on the Ntsr1 + neurons suggest that they are of the distinctive wide-field anatomical type, with a broad dendritic fan (*Major et al., 2000*), and that they have the largest receptive fields in the superficial SC (*Gale and Murphy, 2014*). In fact, we did find very large receptive fields among Ntsr1 + neurons, but surprisingly also many small ones (*Figure 5C*). Notably another recent study also reported a large variation of receptive field sizes in this line (*Hoy et al., 2019*). Perhaps some of the Ntsr1 +neurons have the wide-field morphology but are dominated by just one or a few dendritic inputs.

At the level of functional classification, two features stand out (*Figure 5D*): First the inhibitory neurons are more represented by types in Group 1 (types 1–10, 58%). Second the Rorb + and Tac1 + neurons comprise mostly the functional types of Group 2 (types 11–24, 77% and 82% respectively). Recall that these types respond poorly to the simple chirp stimulus but strongly to moving and expanding objects (*Figure 2A*). Beyond that, there is no one-to-one relationship between genetic labels and functional properties: Each of these genetic markers labels neurons with various functions, and vice versa.

## Transformations from retina to superior colliculus

The superficial layers of the superior colliculus receive direct input from most types of retinal ganglion cells. How is visual information transformed as the superior colliculus processes these signals further? To address this directly, our experimental design included a precise copy of the stimuli used previously to classify and distinguish functional types of retinal ganglion cells (*Baden et al., 2016*). Here, we compare the known retinal ganglion cell response types to those we identified in superior colliculus.

As a first-order analysis, we fitted the chirp and color responses of each SC type with a linear combination of the responses of RGC types (*Figure 6A*, *Figure 6—figure supplement 1A–D*; *Román Rosón et al., 2019*). Under the null hypothesis where each SC type is dominated by one RGC type, this should yield fit weights only along the diagonal. Instead, the optimal fit mixes contributions from many RGC types. While these coefficients cannot be interpreted as representing synaptic connectivity, they nonetheless indicate a broad mixing of retinal inputs at the level of superior colliculus.

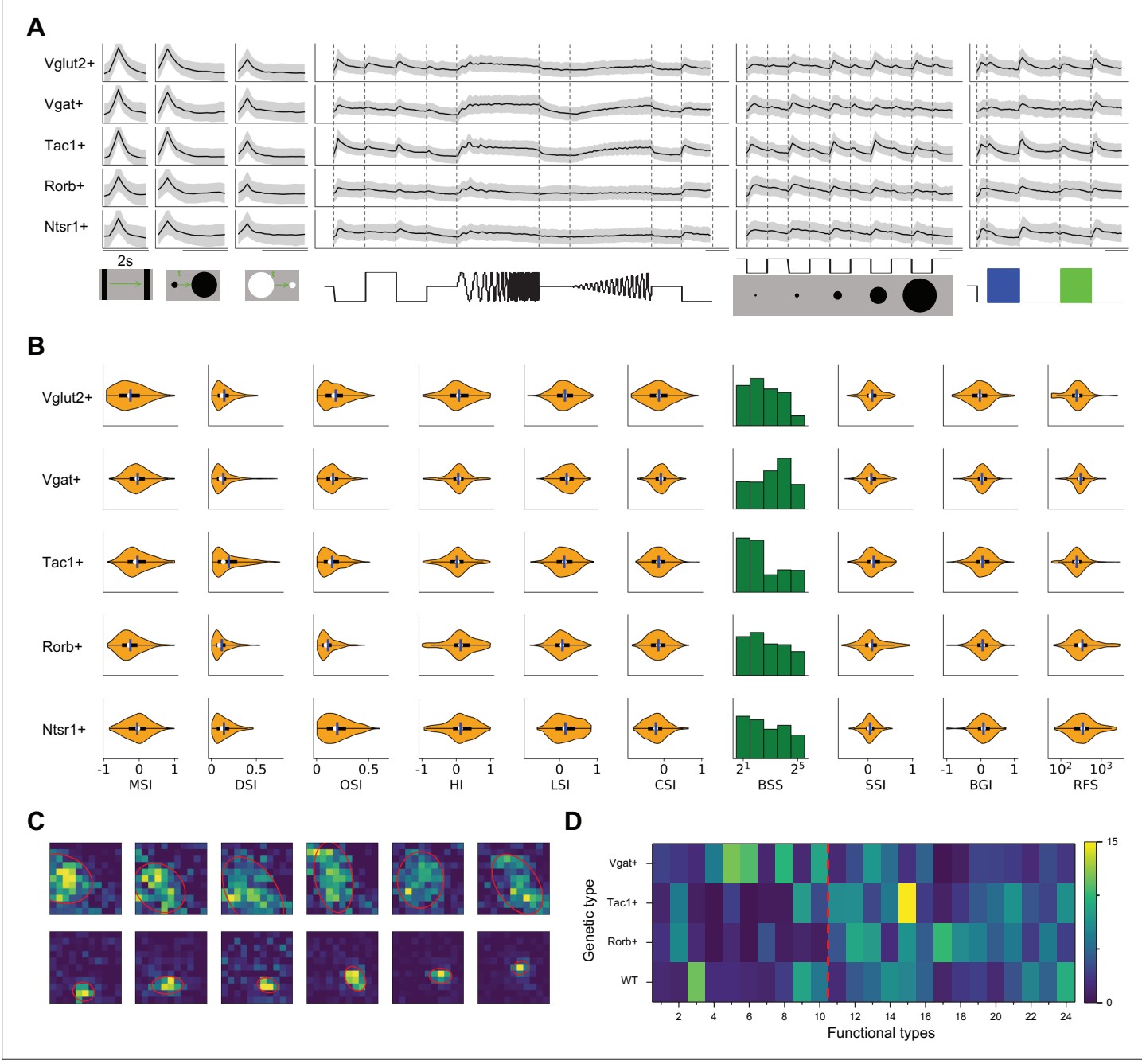

**Figure 5.** Functional properties in genetically labeled populations. (**A**) Average response to the chirp stimulus of five genetically labeled cell types. Display as in *Figure 2A*. (**B**) Functional properties of the genetically labeled types. Display as in *Figure 3A*. (**C**) Example receptive fields of Ntsr1+ neurons mapped with squares flashing Off and fitted with an ellipse. (**D**) The percentage of functional types in mice with different genetic backgrounds. Red dashed line separates Group 1 and Group 2.

A number of interesting features emerge. First, some RGC types have universally excitatory (types 16 (ON DS trans.) and 23 (ON 'mini' alpha)) or universally inhibitory (types 7 (OFF sustained) and 18 (ON transient)) effects on almost all the SC responses. The inhibitory effects could be implemented by recruiting local inhibitory neurons in SC. These RGC types represent a diversity of functional properties. Second, 18 of 32 RGC types have significantly larger contribution to Group 1 (types 1–10) than Group 2 (11–24, $p < 0.05$, two-sample $t$-test). Recall that these groups represent the major division in the dendrogram of SC types (*Figure 2A*).

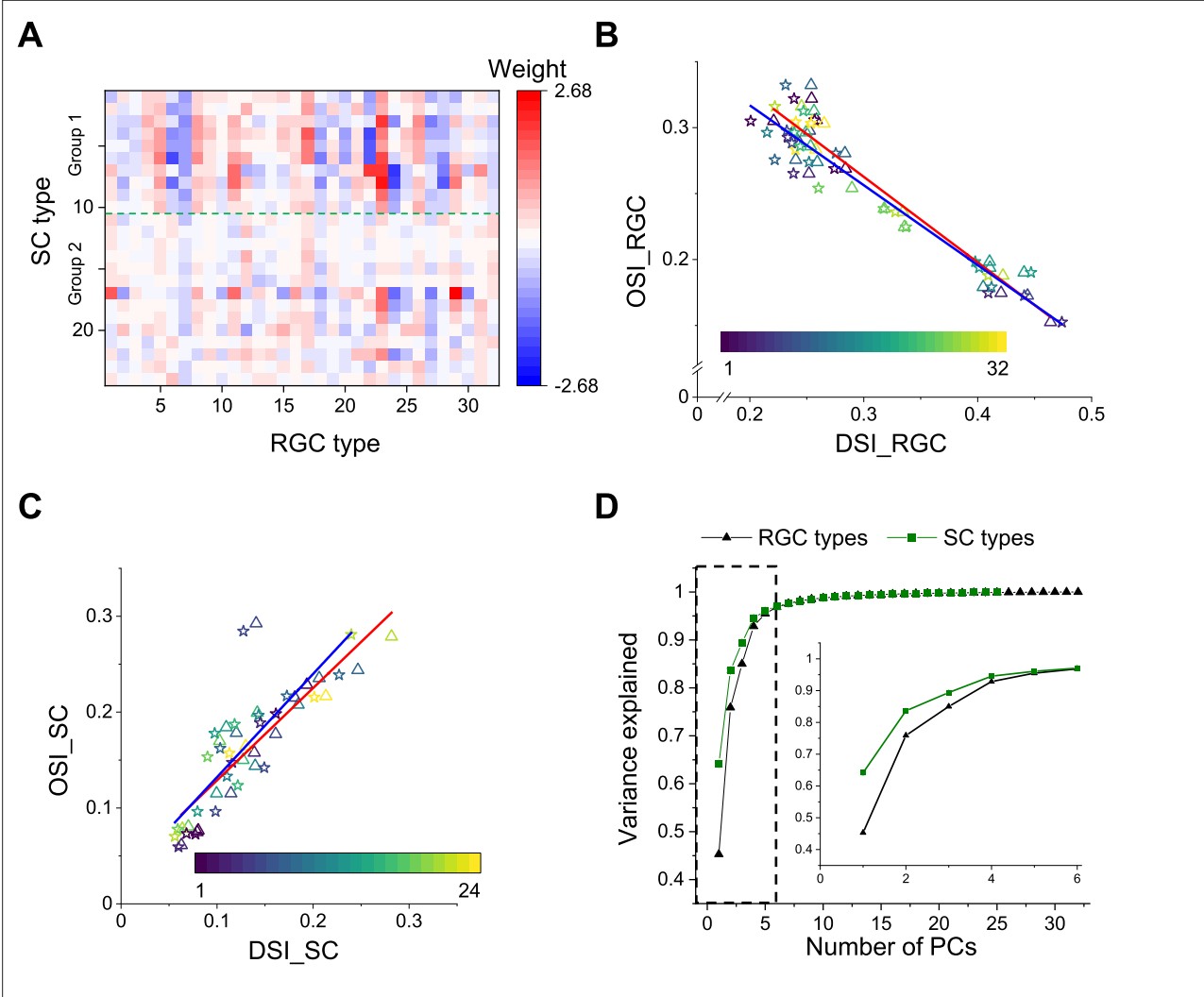

**Figure 6.** Comparison with functional RGC types. (**A**) Weights of different RGC types on each SC type (see Materials and methods). Green dashed line separates Group 1 and Group 2. (**B**) Plot of mean (triangle) and median (star) OSI versus DSI for 32 functional RGC types. *Baden et al., 2016*. Color codes functional type. (**C**) Plot of mean (triangle) and median (star) OSI versus DSI for 24 functional types in the SC. Color codes functional type. (**D**) PCA of the visual responses of SC and RGC types, plotting the fractional explained variance against the number of components. Inset enlarges the part in the dashed rectangle. The uncertainty in these values was <0.01 (SD) in all cases, as estimated from a bootstrap analysis.

The online version of this article includes the following figure supplement(s) for figure 6:

**Figure supplement 1.** Comparisons between SC neurons and RGCs (related to *Figure 6*).

Another substantial transformation from retina to colliculus occurs in the representation of orientation and direction for moving stimuli. In the retina, different ganglion cell types are used to encode the direction vs the orientation of bar stimuli (*Figure 6B*, *Figure 6—figure supplement 1E*). By comparison, in the colliculus it appears that the same cells are selective for orientation and for direction (*Figure 6C*, *Figure 6—figure supplement 1F–I*). Similarly, the retina contains many ganglion cell types with purely On-type or Off-type responses (*Sanes and Masland, 2015*; *Baden et al., 2016*). By comparison, at the level of the colliculus those two pathways are largely combined, and most functional types are of the On-Off type with a bias towards Off responses (*Figure 2A*). These examples suggest a reduction in diversity during transformation from retina to colliculus. To test this more globally, we subjected the entire set of chirp-color responses to a principal component analysis, and compared the results from RGC types to those from SC types (*Figure 6D*). In the superior colliculus, the variance in the visual responses can be explained by fewer principal components than in the

retina. This suggests a reduction in diversity from retina to colliculus, as might be expected from selective visual filtering of features essential to guide the animal's behavior.

## Discussion

Parallel pathways are central to the architecture of biological vision. The visual pathway forks already at the photoreceptor synapse – into ON and OFF representations – and by the time the signal emerges from the eye it has been split into about 40 channels. Each of these pathways is carried by a type of retinal ganglion cell, tuned to certain spatio-temporal features of the visual input, and its neural population covers the entire visual field. How these parallel signals are combined and elaborated in subsequent neural stages of the visual system to sustain the behavioral needs of the animal remains a fundamental question for vision science. Here, we sought to follow these parallel pathways into the superior colliculus, the most important retinal projection target in rodents, which receives input form ~90% of the retinal ganglion cells. The approach was to survey visual response properties across many neurons in the superficial layers of the superior colliculus, using precisely those stimuli that had been instrumental in defining the parallel pathways at the retinal output.

### Main findings

Based on their responses to a broad set of visual stimuli, neurons in the upper SC fall into 24 functional types (*Figures 2 and 3*). At a coarse level, the types form two groups, depending on whether they respond well to simple chirp stimuli (Group 1) or not (Group 2) (*Figure 2A*). Cells in Group 1 dominate the most superficial regions (<100 μm depth), those in Group 2 the deeper levels (*Figure 4H*). Unlike in the retina – where cells of the same type are spaced laterally at a regular distance – in the SC cells of the same type seem to attract each other. In fact many types appear in clusters, about 300–500 μm in size (*Figure 4*). Putative excitatory and inhibitory neurons differ in functional properties: On average the inhibitory neurons have a greater preference for large stimuli and for stimulus motion (*Figure 5*). Compared to retinal ganglion cells, none of the SC types match an RGC type exactly; instead different retinal sensitivities get combined at the level of SC (*Figure 6*). Overall, the visual representation in the superficial SC is somewhat less diverse than in the retina (*Figure 6D*).

### Relation to prior work

Although the superior colliculus receives many parallel channels of input from the retina and the visual cortex, prior work classified only 4–5 types of neurons in the SC (*Wang et al., 2010*; *Gale and Murphy, 2014*; *De Franceschi and Solomon, 2018*). The greater diversity of response types in the present study results in part from a different conceptual approach: Instead of defining cell types based on a combination of morphological, electrophysiological, and functional criteria (*Wang et al., 2010*; *Gale and Murphy, 2014*), we focused on visual response properties. This method acknowledges that neurons with the same shape do not necessarily perform the same function in the circuit. Indeed prior work had found that even among Ntsr1 + neurons that have distinctive wide-field shape there is a great range of response properties (*Hoy et al., 2019*).

Another supporting factor is the much greater number of neurons in the present survey, which allows finer distinctions and statistical assessment of robustness of the functional cell typing. Further, much of the prior work was conducted under anesthesia. This has adverse effects on visual selectivity in the superior colliculus, as pointed out by *De Franceschi and Solomon, 2018*, and thus lowers the number of distinguishable cell types.

At the same time, some of the 24 functional types described here are also evident in prior studies: Types 11 and 15 that show rebounded responses after the frequency-modulated flash are similar to the suppressed-by-contrast cell type (*Ito et al., 2017*). Types 9 and 14 show strong orientation and direction selectivity, as reported in prior work (*Inayat et al., 2015*; *De Franceschi and Solomon, 2018*).

The notion of dividing SC types into two groups is supported by a recent report (*Sibille et al., 2022*). From simultaneous recordings of RGC axons and SC neurons, this study showed that one group of SC neurons receives synaptic inputs from functionally homogeneous RGCs while another group combines more functionally diverse inputs. Clearly, this is a powerful way of assessing direct retino-geniculate synapses. By contrast, our analysis includes indirect effects through interneuron

circuitry, and suggests that most SC neurons end up combining signals from multiple retinal channels (*Figure 6A*).

## Implications for visual processing

Regarding the fate of parallel pathways emerging from the retina, one can contemplate two opposing hypotheses: (1) Further divergence of pathways. In this picture downstream circuits split the visual representation further, yielding more functional types of neurons. Each type responds sparsely only when its favorite feature occurs. (2) Convergence of parallel pathways. In this version downstream circuits begin to narrow the visual representation, computing a few variables from the scene that are useful for controlling behavior, while discarding those bits of information that aren't needed. The present observations favor the second hypothesis, for the following reasons:

First, there are fewer recognizable types in this survey of SC responses (24) than there are in the retinal ganglion cell layer (40) under the same tests. Some comments are in order regarding this comparison. In the retina, the number of RGC types is supported by a convergence of functional, molecular, and anatomical criteria (*Sanes and Masland, 2015*), leaving little ambiguity. By comparison, the proposed number of 24 functional types in SC is open to debate. The statistical criterion we used (*Figure 2B*) is not the only option, and one can envision a coarser split into fewer types. Furthermore, the cell types reported here, as well as those in previous studies (*Wang et al., 2010*; *Gale and Murphy, 2014*; *De Franceschi and Solomon, 2018*), may well lie at different levels of the synaptic network, whereas the RGCs all lie in the same network layer of parallel representation. All these biases point in the same direction: There are very likely fewer SC types than RGC types. Second, certain features of the visual stimulus that were separated into different retinal cell types appear combined within the superior colliculus. Notably this is the case for the representation of orientation and motion direction (*Figure 6C*) and for that of On- and Off-signals (*Figure 2A*). Third, a global measure of dimensionality in the neural representation is smaller among SC types than RGC types (*Figure 6D*). Note that ~10% of RGCs avoid the SC, including SPIG1 + cells in the pan-ventronasal retina (*Yonehara et al., 2008*), which could contribute to the reduction of dimensionality.

All this suggests that the functional diversity in the visual pathway may be greatest at the retinal output. Subsequent circuits may act primarily as a switchboard to distribute these RGC signals to different brain targets. For example, a recent study traced the projections from SC to two different target areas, and found that the corresponding SC cells combine inputs from different subsets of retinal ganglion cells (*Reinhard et al., 2019*). It will be interesting to explore how these projection-defined types correspond to the functional types reported here and their putative RGC complements (*Figure 6A*).

Another output target from the superficial SC are the deeper layers, where visual information is combined with other sensory pathways as well as motor signals. These are out of reach for effective optical recording but can be accessed by electrodes. There one encounters response properties not seen in the retina, foremost among them a pronounced habituation to repeated stimuli (*Dräger and Hubel, 1975*; *Horn and Hill, 1966*; *Lee et al., 2020*). Also, the range of response properties narrows systematically with depth, reflecting an increased selectivity for behaviorally-relevant visual features (*Lee et al., 2020*). Similarly, it is possible that the functional types varies across regions of the SC. Stimulating the region that represents the upper or low visual field elicits avoidance or orientating behavior respectively (*Sahibzada et al., 1986*), suggesting that other functional types relevant to orienting behaviors likely exist in regions outside the posterior medial sector studied here.

## The spatial organization of functional cell types

The patchy organization of functional cell types (*Figure 4*) extends the results reported previously regarding specific neuronal response properties, notably the preferred orientation (*Feinberg and Meister, 2015*; *Ahmadlou and Heimel, 2015*; *Sibille et al., 2022*) or movement direction (*Li et al., 2020*; *de Malmazet et al., 2018*). A common theme to all these reports is that the rules of visual processing seem to vary across broad regions of the visual field (for a dissenting report see *Chen et al., 2021*). In the present study we found that different regions of the visual field, measuring ~20–40 degrees across, are covered by a different complement of functional types. Our recordings were limited to the posterior-medial portion of the SC (upper temporal visual field), and thus the

global organization of these functional types remains unclear. Also, it is possible that additional functional types will emerge in other parts of the visual field.

It is important to contrast this organization with inhomogeneities found elsewhere in the visual system. Among retinal ganglion cells (RGCs), the mosaic arrangement of neurons within a type guarantees that they are properly spaced, such that each point in the visual field is handled by at least one such ganglion cell. Within a given RGC type, the response properties may vary gradually across the visual field, typically in a naso-temporal or dorso-ventral gradient (*Bleckert et al., 2014*; *Warwick et al., 2018*). This is very different from the regional specializations encountered in the SC. In the primary visual cortex of large animals, one often finds that functional types are organized in patches or stripes. However, the scale of this functional anatomy is finer than the receptive field size: This guarantees again that each point in the visual field is handled by neurons of every type (*Blasdel and Campbell, 2001*). By contrast, in the SC the observed patches are 10–20 times larger than receptive fields, which implies a regional specialization of certain visual processes. The reasons for this specialization remain unclear. One can certainly invoke ecological arguments for treating the upper and lower visual fields differently, but on the scale of 30 degrees the purpose is less obvious.

Future work may also inspect the downstream effects of this patchy organization. One hypothesis is that a given functional type gathers visual information for distribution to one of the many downstream visual centers. If so, then a retrograde tracing from one of the SC's target regions should cover only a patch of the visual field. The rapid progress in connectomic and transcriptomic methods for mapping cell types promises further insights into this unusual functional organization.

## Materials and methods
### Animal
Laboratory mice of both sexes were used at age 2–4 months. The strains were C57BL/6J (wildtype), Vglut2-ires-Cre (B6J.129S6(FVB)-*Slc17a6^{tm2(cre)Lowl}*/MwarJ, JAX: 028863), Vgat-ires-Cre (B6J.129S6(FVB)-*Slc32a1^{tm2(cre)Lowl}*/MwarJ, JAX: 028862), Tac1-IRES2-Cre-D (B6;129S-*Tac1^{tm1.1(cre)Hze}*/J, JAX: 021877) (*Harris et al., 2014*), Rorb-IRES2-Cre-D (B6;129S-*Rorb^{tm1.1(cre)Hze}*/J, JAX: 023526) (*Harris et al., 2014*), and Ntsr1–GN209–Cre (Genset: 030780-UCD) (*Gerfen et al., 2013*). Vglut2-ires-Cre and Vgat-ires-Cre mice express Cre recombinase in *Vglut2*-expressing and *Vgat*-expressing neurons, respectively. Tac1-IRES2-Cre-D and Rorb-IRES2-Cre-D mice express Cre recombinase in *Tac1*-expressing and *Rorb*-expressing neurons, respectively. Cre-D indicates neo/hydro is deleted. Ntsr1–GN209–Cre mice express Cre recombinase in *Ntsr1-GN209*-expressing neurons. GN209 is the founder line. All animal procedures were performed according to relevant guidelines and approved by the Caltech IACUC.

### Viral injection
We injected adeno-associated virus (AAV) expressing non-floxed GCaMP6 (AAV2/1.hSyn1.GCaMP6f.WPRE.SV40) into the SC of wild-type mice (C57BL/6 J), and AAV expressing floxed GCaMP6 (AAV1.Syn.Flex.GCaMP6f.WPRE.SV40) into the SC of Vglut2-ires-Cre, Vgat-ires-Cre, Tac1-IRES2-Cre-D, Rorb-IRES2-Cre-D, and Ntsr1–GN209–Cre mice. After 2–3 weeks, we implanted a cranial window coupled to a transparent silicone plug that rested on the surface of the SC and exposed its posterior-medial portion. This portion of the SC corresponds to the upper-temporal part of the visual field. The optics remained clear for several months, which enabled long-term monitoring of the same neurons. Two-photon microscopy was used to image calcium signals in the SC of head-fixed awake mice 3 weeks to 2 months after viral injection.

### In vivo two-photon calcium imaging
For imaging experiments, the animal was fitted with a head bar, and head-fixed while resting on a rotating treadmill. The animal was awake and free to move on the treadmill, but not engaged in any conditioned behavior. Two-photon imaging was performed on a custom-built microscope with a 16×, 0.8 NA, 3 mm WD objective (Nikon). A Ti:Sapphire laser with mode-locking technique (Spectra-Physics Mai Tai HP DeepSee) was scanned by galvanometers (Cambridge). GCaMP6f was excited at 920 nm and laser power at the sample plane was typically 20 − 80 mW. A 600 μm × 600 μm field of view was scanned at 4.8 Hz as a series of 250 pixel × 250 pixel images and the imaging depth was

up to 350 μm. Emitted light was collected with a T600/200dcrb dichroic (Chroma), passed through a HQ575/250 m-2p bandpass filter (Chroma), and detected by a photomultiplier tube (R3896, Hamamatsu). Artifacts of the strobed stimulus (see below) were eliminated by discarding 8 pixels on either end of each line. The animal's locomotion on the treadmill and its pupil positions were recorded and synchronized to the image acquisition. The head-fixed animal performs only rare eye movements and locomotion (*Li et al., 2020*).

## Visual stimulation

An LCD screen with LED backlight was placed 18 cm away from the mouse's right eye. The center of the monitor was at 95° azimuth and 25° elevation in the laboratory frame, and the monitor covered a visual field of 106° × 79°. The visual angle that covers the receptive fields of recorded neurons is 60°–140° azimuth and 0°–50° elevation (*Figure 4—figure supplement 2*). The monitor's LED illuminator was strobed for 12 μs at the end of each laser scan line to minimize interference of the stimulus with fluorescence detection. The monitor was gamma-corrected. For measuring the functional properties, we presented six types of visual stimuli. (1) A full-field moving black bar (5° width at 50°/s) in 12 directions to measure the orientation selectivity and direction selectivity. The sequence of directions was pseudo-randomized. (2) An expanding black disc (diameter 2° to 60° at a speed of 60°/s, stationary at 60° for 0.25 s, followed by grey background for 2 s) and a receding white disc (60° to 2° at a speed of 60°/s, other parameters same as expanding disc) to measure looming-related responses. (3) Sparse (one at a time) 5° × 5° flashing squares (11×11 squares, 1 s black or white +1 s grey) to map the receptive field (RF); (4) A 10° × 10° square modulated by a "chirp" in frequency or amplitude (3 s black +3 s white +3 s black +3 s grey +8 s frequency modulation ($2^{-1:3}$ Hz)+3 s grey +8 s amplitude modulation (0 : 1)+3 s grey +3 s black) centered on the RF to measure temporal properties (*Baden et al., 2016*); (5) A 10° × 10° square flashing blue or green (1 s black +3 s blue +4 s black +3 s green +3 s black) centered on the RF to measure the color preference; (6) A flashing disc (2 s black +2 s grey) with different size (2°, 4°, 8°, 16°, 32°) centered on the RF to measure the size tuning. All stimuli were displayed for 10 repetitions. Stimuli of types 4, 5, and 6 were also repeated identically at locations in a 3×3 array shifted by 10°. This effectively covered the visual field recorded during a given imaging session. For each neuron, we based the analysis on the stimulus closest to its receptive field center.

## Analysis of calcium responses
### Measurement of calcium responses

Brain motion during imaging was corrected using SIMA (*Kaifosh et al., 2014*) or NoRMCorre (*Pnevmatikakis and Giovannucci, 2017*). Regions of interest (ROIs) were drawn manually using Cell Magic Wand Tool (ImageJ) and fitted with an ellipse in MATLAB. Fluorescence traces of each ROI were extracted after estimating and removing contamination from surrounding neuropil signals as described previously (*Feinberg and Meister, 2015*; *Li et al., 2020*; *Göbel and Helmchen, 2007*; *Kerlin et al., 2010*). The true fluorescence signal of a neuron is $F_{\text{true}} = F_{\text{raw}} - (r \cdot F_{\text{neuropil}})$, where $r$ is the out-of-focus neuropil contamination factor and the estimated value for our setup is ~ 0.7. Slow baseline fluctuations were removed by subtracting the eighth percentile value from a 15 s window centered on each frame (*Dombeck et al., 2007*).

For any given stimulus, the response of a neuron was defined by the fluorescence trace in its ROI during the stimulus period:

$$R = \frac{F - F_0}{F_0} \tag{1}$$

where $F$ is the instantaneous fluorescence intensity and $F_0$ is the mean fluorescence intensity without visual stimulation (grey screen).

Two criteria were applied to interpret ROIs as neurons: (1) The size of the ROI was limited to 10–20 μm to match the size of a neuron; (2) The response from the ROI had to pass a signal-to-noise ratio (SNR) of 0.35 (*Baden et al., 2016*):

$$SNR = \frac{\text{Var}[\langle C \rangle_r]_t}{\langle \text{Var}[C]_r \rangle_t} \tag{2}$$

where $C$ is the $N_t$ (time samples) $\times N_r$ (stimulus repetitions) response matrix, $t = 1, \ldots, N_t$ and $r = 1, \ldots, N_r$, $\langle \cdot \rangle_r$ and $\langle \cdot \rangle_t$ are the means over repetitions or time respectively, and $\mathrm{Var}[\cdot]_r$ and $\mathrm{Var}[\cdot]_t$ are the corresponding variances. All ROIs meeting these criteria were selected for further analysis, yielding a total of 3414 neurons, including 490 neurons from four wild type mice, 337 neurons from one Vglut2-ires-Cre mouse, 1085 neurons from three Vgat-ires-Cre mice, 720 neurons from three Tac1-IRES2-Cre-D mice, 485 neurons from four Rorb-IRES2-Cre-D mice, and 297 neurons from one Ntsr1–GN209–Cre mouse.

## Quantification of functional properties

The functional properties introduced in *Figure 3* are defined as follows:

The response to motion (RtM) is the response value during moving-bar stimuli with the largest absolute value. For neurons suppressed by motion this will be negative.

To quantify the tuning of a neuron to motion directions, we calculated the direction selectivity index (DSI) as the normalized amplitude of the response-weighted vector sum of all directions:

$$DSI = \frac{\left| \sum_k R(\rho_k) \times e^{i\rho_k} \right|}{\sum_k R(\rho_k)} \tag{3}$$

where $\rho_k$ is the $k^{\mathrm{th}}$ direction in radians and $R(\rho_k)$ is the peak response at that direction.

To quantify the orientation tuning, we calculated the orientation selectivity index (OSI) as the normalized amplitude of the response-weighted vector sum of all orientations:

$$OSI = \frac{\left| \sum_k R(\theta_k) \times e^{2i\theta_k} \right|}{\sum_k R(\theta_k)} \tag{4}$$

where $\theta_k$ is the $k^{\mathrm{th}}$ orientation in radians and $R(\theta_k)$ is the peak response at that orientation.

To quantify the habituation to the expanding black disc, we calculated the habituation index (HI):

$$HI = \frac{R_1 - R_{10}}{R_1 + R_{10}} \tag{5}$$

where $R_1$ and $R_{10}$ are the peak response to the first and the tenth looming stimulus respectively.

To quantify the selectivity to the expanding black disc over the receding white disc, we calculated the looming selectivity index (LSI):

$$LSI = \frac{R_{\mathrm{k}} - R_{\mathrm{w}}}{R_{\mathrm{k}} + R_{\mathrm{w}}} \tag{6}$$

where $R_{\mathrm{k}}$ is the peak response to the black expanding disc and $R_{\mathrm{w}}$ is the peak response to the white receding disc.

To quantify the selectivity to moving stimuli over the flashing stimuli, we calculated the motion selectivity index (MSI):

$$MSI = \frac{R_{\mathrm{m}} - R_{\mathrm{f}}}{R_{\mathrm{m}} + R_{\mathrm{f}}} \tag{7}$$

where $R_{\mathrm{m}}$ is the peak response to the moving bar at the preferred direction and $R_{\mathrm{f}}$ is the peak response to the flashing chirp stimulus.

To quantify the selectivity to On/Off contrast, we calculated the contrast selectivity index (CSI):

$$CSI = \frac{R_{\mathrm{On}} - R_{\mathrm{Off}}}{R_{\mathrm{On}} + R_{\mathrm{Off}}} \tag{8}$$

where $R_{\mathrm{On}}$ is the peak response to the flashing white square and $R_{\mathrm{Off}}$ is the peak response to the flashing black square.

To quantify whether neurons show transient or sustained responses to flash stimuli, we calculated the peak-final selectivity index (PFSI):

$$PFSI = \frac{R_{\mathrm{peak}} - R_{\mathrm{final}}}{R_{\mathrm{peak}} + R_{\mathrm{final}}} \tag{9}$$

where $R_{\mathrm{peak}}$ is the peak response to the flashing white/black square that elicited larger responses, and $R_{\mathrm{final}}$ is the final response to that stimulus.

The quantify the selectivity to the flash frequency, we calculated the frequency selectivity index (FSI):

$$FSI = \frac{R_{\text{low}} - R_{\text{high}}}{R_{\text{low}} + R_{\text{high}}} \tag{10}$$

where $R_{\text{low}}$ is the peak response in the first 3 s to the flashing frequency modulation, and $R_{\text{high}}$ is the peak response in the last 2 s to the frequency modulation.

We measured the response after frequency modulation (RaFM) as the difference between the response amplitude at 1.6 s after the stop of the frequency modulation and the baseline. Similarly, the response after amplitude modulation (RaAM) was measured as the difference between the response amplitude at 1.6 s after the stop of the amplitude modulation and the baseline.

The best stimulation size (BSS) was defined the size of flashing black disc that elicited the largest responses.

To quantify the surround suppression, we calculated the surround suppression index (SSI):

$$SSI = \frac{R_{\text{small}} - R_{\text{large}}}{R_{\text{small}} + R_{\text{large}}} \tag{11}$$

where $R_{\text{small}}$ is the peak response to the flashing black disc with a diameter of 2 degrees, and $R_{\text{large}}$ is the peak response to the flashing black disc with a diameter of 32 degrees.

To quantify the color preference, we calculated the blue-green index (BGI):

$$BGI = \frac{R_{\text{b}} - R_{\text{g}}}{R_{\text{b}} + R_{\text{g}}} \tag{12}$$

where $R_{\text{b}}$ is the response to the flashing blue stimulus and $R_{\text{g}}$ is the response to the flashing green stimulus.

To quantify the receptive field size (RFS), the calcium responses at 11×11 locations were fitted with a 2-D Gaussian function (*Equation 13*):

$$f = A \cdot e^{-\frac{((x-E)\cos(D) - (y-F)\sin(D))^2}{2B^2} - \frac{((x-E)\sin(D) + (y-F)\cos(D))^2}{2C^2}} + G \tag{13}$$

The RF size is defined as the area at the tenth of maximum, which equals $\pi \cdot 2\ln 10 \cdot B \cdot C$. We omitted analysis of the RF if the coefficient of determination for this fit was below 0.5 (*Figure 3—figure supplement 1I*). The RF size of neurons with the coefficient of determination larger than 0.5 is shown in *Figures 3–5*.

## Construction of the feature matrix

We reduced the dimensionality of the calcium response traces, by approximating them with a weighted sum of features, while requiring that the weight coefficients be sparse. For this analysis we included the neuronal responses to moving bars (MB), expanding black and receding white disc (EBD and RWD), chirp, color, flashed black discs with different sizes (FDDS). Neuronal responses to MB, EBD, and RWD were aligned to the peak or the trough to remove the effect of RF position. We focused on neurons which responded robustly (SNR >0.35) to at least one stimulus. For each neuron and each stimulus, the response was normalized to [0,1]. The optimal features were extracted with sparse principal components analysis (spca; *Mairal et al., 2009*), as implemented in the scikit-learn package (*Equation 14*):

$$\mathbf{D} = \mathbf{XF} \tag{14}$$

where $\mathbf{D}$ ($N_{\text{neuron}} \times N_{\text{time}}$) is the data matrix denoting neuronal responses to a visual stimulus, $\mathbf{F}$ ($N_{\text{feature}} \times N_{\text{time}}$) is the matrix with the time course corresponding to each feature, and $\mathbf{X}$ ($N_{\text{neuron}} \times N_{\text{feature}}$) is the matrix of weight coefficients. $\mathbf{F}$ is regularized so that only sparse values in each row are non-zero, and $\mathbf{FF}^{\top} \approx \mathbf{I}$.

We extracted 6 features from the responses to MB at the preferred direction, 6 features from the responses to EBD and RWD, 20 features from the chirp, 8 features from color stimuli, and 10 features from FDDS (*Figure 2—figure supplement 1A*). These extracted features were combined with HI, DSI, OSI, and MSI to make the feature matrix. Each feature was normalized so that the mean is 0 and the standard deviation is 1.

## Clustering of the feature matrix

We used a Gaussian mixture model (GMM) to fit the distribution of neurons in the space of features:

$$p(x) = \sum_{i=1}^{K} \phi_i N(x|\mu_i, \sigma_i) \tag{15}$$

$$N(x|\mu_i, \sigma_i) = \frac{1}{\sigma_i \sqrt{2\pi}} \exp\left(-\frac{(x-\mu_i)^2}{2\sigma_i^2}\right) \tag{16}$$

$$\sum_{i=1}^{K} \phi_i = 1 \tag{17}$$

where $p(x)$ is the probability density of the feature vector $x$, $K$ is the number of component Gaussian functions, and $\phi_i$ is the weight for $i^{\text{th}}$ Gaussian function $N(x|\mu_i, \sigma_i)$ in the feature space. We optimized the parameters using the EM algorithm (sklearn.mixture.GaussianMixture in the package scikit-learn). We varied the number of components from 2 to 50 and evaluated the quality with the Bayesian information criterion (BIC) (*Kass and Raftery, 1995*):

$$BIC = -2 \ln L + k \ln n \tag{18}$$

where $L = p(x|\theta, M)$, is the maximized likelihood of model $M$, $x$ is the observed data, $\theta$ are the parameters that maximize the likelihood, $k$ is the number of parameters in the model, $n$ is the number of neurons. For each putative number of components (2–50), we performed the EM fit starting from 1000 random initial states, and chose the fit with the smallest BIC. This minimal BIC is plotted against the number of components in *Figure 2B*.

To evaluate the stability of clusters, we applied sub-sampling analysis (*Hennig, 2007*). We randomly sub-sampled 90% of the dataset 1000 times and fitted the subset with a GMM using the best cluster number determined from the full dataset. For each original cluster, we calculated its Jaccard similarity coefficient (JSC) with the subsets:

$$JSC = \frac{1}{N} \sum_{i=1}^{N} \max_j \left\{ \frac{\left| C_{\text{full}} \cup C_{\text{sub}}^j \right|}{\left| C_{\text{full}} \cap C_{\text{sub}}^j \right|} \right\} \tag{19}$$

where $N$ is the number of subsets, $C_{full}$ is the cluster in the full dataset, and $C_{\text{sub}}^j$ is the th cluster in one subset. Clusters with JSC below 0.5 were considered unstable. The unstable cluster was merged with the cluster that had the highest between-cluster rate if that rate was > 35% (*Gouwens et al., 2019*); otherwise, it was marked as unstable in the figure.

To assess the robustness of classification by the EM algorithm, we measured the probability that a pair of cells is classified into the same cluster in different subsets, and calculated the co-association matrix (*Fred and Jain, 2005*):

$$CAM(i,j) = \frac{n_{i,j}}{N} \tag{20}$$

where $n_{i,j}$ is the number of times that the pair $(i,j)$ is assigned to the same cluster in $N$ attempts. The between-cluster rate is defined as the cluster-wise average of the co-association matrix.

To plot the dendrogram, we applied a linkage algorithm (scipy.cluster.hierarchy.linkage) to the means of different clusters in the feature space. We measured the Euclidean distance between two points and defined the distance between two clusters with Ward's minimum variance method.

## Relative selectivity index

Relative selectivity index (RSI) is defined as the difference of functional property between one type and a reference number:

$$RSI(i,j) = F_{i,j} - F_{i,ref} \tag{21}$$

where $F_{i,j}$ is functional property $i$ of type $j$ and $F_{i,ref}$ is the reference of functional property $i$. The functional properties are response to motion (RtM), motion selectivity index (MSI), direction selectivity index (DSI), orientation selectivity index (OSI), habituation index (HI), looming selectivity index (LSI), contrast selectivity index (CSI), peak-final selectivity index (PFSI), response after frequency modulation (RaFM), response after amplitude modulation (RaAM), best stimulation size (BSS), surround

suppression index (SSI), blue-green selectivity index (BGSI), receptive field size (RFS). The reference numbers are RtM: 0, DSI: 0.15, OSI: 0.15, HI: 0, LSI: 0, MSI: 0, CSI: 0, PFSI: 0.5, FSI: 0, RaFM: 0, RaAM: 0, BSS: $2^3$, SSI: 0, BGI: 0, RFS: $10^{2.46}$.

## Analysis of the anatomical arrangement of functional cell types

For the results on anatomical arrangement (*Figure 4*), only recording sessions with >5 neurons in a field of view were included. The density recovery profile (DRP) plots the probability per unit area of finding a cell as a function of distance from a cell of the same type (*Rodieck, 1991*). We first defined the region of interest (ROI) as the convex hull of all neurons in an image. Within this ROI, we measured the distances from each reference cell to all of the other cells and histogrammed those, which yields

$$N(r)\Delta r = \text{average number of cells at radii between } r \text{ and } r + \Delta r \tag{22}$$

Then we measured the average area $A(r)\Delta r$ at distance between $r$ and $r + \Delta r$ from any reference point in the window.

Finally the DRP was calculated as

$$\rho(r) = \frac{N(r)}{A(r)} \tag{23}$$

The density ratio (DR) for each type is calculated as

$$DR(i) = \frac{\rho_i(r_0)}{\rho_i(r_1)} \tag{24}$$

where $\rho_i(r_0)$ is the mean density of functional type $i$ within $0.5R$ of a cell of that type, $R$ is the average receptive field diameter of that type, and $\rho_i(r_1)$ is the mean density in the annulus spanning $0.5 - 1R$. To connect anatomical distance in the SC with angular distance in the visual field we assumed that 1 mm corresponds to 88 degrees (*Dräger and Hubel, 1976*).

To quantify how the functional type of one neuron is related to the functional types of its neighbors, we calculated the density of different types:

$$D(i,j) = \frac{n_{i,j}}{N_j} \tag{25}$$

where $n_{i,j}$ is number of neurons of functional type $j$ within a certain distance to a neuron of type $i$, and $N_j$ is the number of neurons of functional type $j$ in the same area if neurons were uniformly distributed.

To quantify the relationship between two types of neurons, we calculated their normalized distance (ND) for each image:

$$ND(i,j) = \frac{d_{i,j}}{0.5 \times (pd_i + pd_j)} \tag{26}$$

where $d_{i,j}$ is the Euclidean distance between the center of types $i$ and $j$, and $pd_i$ is the mean pairwise distance between neurons of type $i$.

To quantify the significance of the separation between two types, we shuffled labels for all neurons in these two clusters and calculated the p-value with bootstrap analysis to test whether the two types are significantly separated. If the maximum p-value of all images that have $\geq 10$ neurons for both types is $\leq 0.01$, these two types are significantly separated.

## Retina-SC transformation

For each type of SC neuron, we asked whether its responses can be explained by superposition of a small number of retinal ganglion cell types (*Figure 6A*). Given the known responses of RGC types to these same stimuli (*Baden et al., 2016*) we approximated the response of each SC type as a weighted combination of RGC responses:

$$\arg\min_{\mathbf{a}} \|\mathbf{Xa} - \mathbf{y}\|_2 \tag{27}$$

where $\mathbf{y}$ is the response vector of the SC type, $\mathbf{X}$ is the matrix of the response vectors to the same stimuli for all types of RGCs, and $\mathbf{a}$ is the desired set of weights. The prediction error is quantified as

$$\frac{\|\mathbf{Xa} - \mathbf{y}\|_2}{\|\mathbf{y}\|_2} \tag{28}$$

## Acknowledgements

MM was supported by grants from NIH (R01 NS111477) and from the Simons Foundation (543015SPI). Y-t L was supported by a grant from the NEI (K99EY028640) and a Helen Hay Whitney Postdoctoral Fellowship.

## Additional information

### Competing interests

Markus Meister: Reviewing editor, eLife. The other author declares that no competing interests exist.

### Funding

| Funder | Grant reference number | Author |
|---|---|---|
| National Institute of Neurological Disorders and Stroke | R01 NS111477 | Markus Meister |
| Simons Foundation | 543015SPI | Markus Meister |
| National Eye Institute | K99EY028640 | Ya-tang Li |
| Helen Hay Whitney Foundation | | Ya-tang Li |

The funders had no role in study design, data collection and interpretation, or the decision to submit the work for publication.

### Author contributions

Ya-tang Li, Conceptualization, Software, Formal analysis, Funding acquisition, Validation, Investigation, Methodology, Writing – original draft, Writing – review and editing; Markus Meister, Conceptualization, Supervision, Funding acquisition, Validation, Writing – original draft, Writing – review and editing

### Author ORCIDs

Ya-tang Li http://orcid.org/0000-0003-2763-1534
Markus Meister http://orcid.org/0000-0003-2136-6506

### Ethics

All animal procedures were performed according to relevant guidelines and approved by the Caltech IACUC (protocol 1656).

### Decision letter and Author response

Decision letter https://doi.org/10.7554/eLife.82367.sa1
Author response https://doi.org/10.7554/eLife.82367.sa2

## Additional files

### Supplementary files

• MDAR checklist

### Data availability

Data and code are available in a Caltech DATA Repository (https://doi.org/10.22002/w3n8w-wgx37) and in a public Github repository (https://github.com/yatangli/Li-CellTypes-2023 copy archived at *Li and Meister, 2023*).

The following dataset was generated:

| Author(s) | Year | Dataset title | Dataset URL | Database and Identifier |
|---|---|---|---|---|
| Y-T Li | 2023 | Code and data for Li & Meister (2023) Functional Cell Types in the Mouse Superior Colliculus | https://doi.org/10.22002/w3n8w-wgx37 | Caltech DATA, 10.22002/w3n8w-wgx37 |

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
