## [Editor Report]

This important paper will be of interest to neuroscientists interested in how the representation of sensory information is refined between the sensory periphery and more central areas. The work provides compelling evidence for a much greater diversity of functional cell types in the superior colliculus than previously suggested, and that the functional organization of cell types in the superior colliculus is distinct from that of the retina.

---

## [Decision Letter]

**Decision letter after peer review:**

Thank you for submitting your article "Functional Cell Types in the Mouse Superior Colliculus" for consideration by *eLife*. Your article has been reviewed by 3 peer reviewers, including Fred Rieke as the Reviewing Editor and Reviewer #1, and the evaluation has been overseen by a Reviewing Editor and Claude Desplan as the Senior Editor.

Essential revisions:

The reviewers all agreed that the data presented in the paper are quite valuable, but that several issues need to be improved before the paper can be fully evaluated. Chief among these are:

1. Results from the genetic lines are not integrated well with the rest of the paper, and this misses an opportunity to provide more information about the properties of the labeled cells and how they fit into the remainder of the recorded population.

2. Some key statements lack confirming analyses – e.g. the separation of cells into those that respond to the chirp stimulus and those that do not.

3. Related to the point above, the paper contains several general statements about how SC is organized (e.g. the number of cell types and spatial organization of these types). The paper would benefit from focusing on those that are directly supported by the results and appropriate analyses. It is not entirely clear which those are without seeing how additional analyses turn out.

These and other important points are detailed in the individual reviews below.

*Reviewer #1 (Recommendations for the authors):*

In some parts of the manuscript, it is unclear how the conclusions were reached. An example is l. 93-99, where the authors say that "Group 1 … is distinct from Group 2 … in that it responds more strongly to the chirp stimuli", but this statement is not accompanied by any quantification. "Almost all these types are excited by both On and Off stimuli" – do the authors refer to the responses to the steps in the chirp or any other part of the data? No quantification / statistics are given for the statements regarding the differential sensitivity to temporal frequency. Similar in l. 107 ff, where statements about strong correlations are made, but not quantified. Or in l. 203, where it is stated that the sign of contribution changes between groups 1 and 2. Please add such information here and throughout.

Several parts of the manuscript are directly related to the recently published findings from the Kremkow lab (10.1038/s41467-022-32775-2). While I assume that the questions have evolved independently in the two labs and that the data have been collected in parallel, I still find it somewhat disappointing that the important work from a junior π is not cited in the manuscript. Beyond acknowledging the achievement, it would be worth discussing the similarities and differences between the two studies, which I perceive as clearly complementary. Similarly, I think the Baden dataset deserves a citation also in text around l. 196 (not only in the figure legend), and the idea of fitting higher visual responses for a weighted sum of RGC inputs was previously performed for dLGN in Roman-Roson et al., 2019.

The authors use an elegant approach of imaging SC while leaving cortex intact, but this comes with the caveat that only a small portion of SC is accessible. The authors touch upon this point, but should discuss in more depth to which degree some of the features they observe might or might not be related to this caveat.

I think that an asset of the paper is the imaging of cre mouse lines, but I also have a number of questions. While the authors stated that in the overall population, they have mostly responses to on and off, they find off-type responses in some of the cre lines (e.g., l. 173). Why is this the case? Can the authors rule out that neurons with on or off responses are mixed in the clusters presented in Figure 2, and thus the cluster response is on-off? Also, instead of saying that the number of neurons is too low for clustering the data from the cre-lines, could the authors try to find the best-matching cluster of the 24 clusters obtained in the large population of neurons?

*Reviewer #2 (Recommendations for the authors):*

Li and Meister recorded the visual responses of more than 5000 neurons from the visual layers of the superior colliculus of 16 awake head-fixed mice allowed to run on a disc. Clustering of the visual responses suggests a greater diversity of functional cell types (24) in the superior colliculus than previously suggested (5 – 10). Neurons fell into two groups, those that respond similarly to retinal ganglion cells and a second group that showed more specialized stimulus selectivity. By comparing this data set with comparable recordings from the mouse retina (Baden et al., 2016) the authors suggest two differences: (i) The visual representation within the superior colliculus has a lower dimensionality than in the retina, (ii) In contrast to the retina, neurons of the same functional type form local clusters. This dataset has the potential to add important details to our understanding of how visual response properties are organized within the superior colliculus and highlight some potentially important key differences with the retina.

While this paper has collected impressive data, there are two significant weaknesses that limit the impact of this work:

1. Clear information about the relationship between the functional classes and genetically labeled populations.

2. Clear description of how the different functional cell types are distributed across visual space.

Providing a link to genetically targetable neuronal components of the superior colliculus would dramatically increase the usability of the data while describing how the set of functional cell types is distributed across the visual field would provide biological insight into the functional organization of the mouse superior colliculus.

1. The integration of the data from the different genetically targeted cell types into the complete data set is not explicit or transparent. As currently presented it is unclear if this is exclusively an issue of analysis or a limitation of the data. An understanding of how functional classes are represented within and among different genetic cell types would make the dataset much more useful to the community and help to better assess the utility of these mouse lines. We have the following questions and requests.

1.1. Please provide information about the distribution of the 24 functional classes for each genetic cell type.

1.2. How separate are the different genetic cell classes?

1.3. Is the functional segregation/overlap consistent with the anatomical overlap of these cell types (e.g Ntsr, tac1, and VGlut2, Zhou et al., 2017; Vgat and *Rorb*, Gale and Murphy, 2018)?

1.4. What is the degree of anatomic overlap between genetic lines?

1.5. Does this predict the amount of functional overlap?

1.6. Please provide the anatomic distribution of the different genetic lines within the superficial layers of the colliculus.

1.7. Please report the number of cells and animals recorded for each cell type in Figure 4.

2. A second issue in the paper is that the distribution of the different functional cell types in visual space is not analyzed. While it is reported in Figure 10 this representation does not report a systematic or unified analysis. In Figure 10 it is suggested that the recording technique measures from ~ 50 – 140 degrees of visual space horizontally and -10 to +60 degrees of visual space vertically. This appears to offer the opportunity to report useful information.

2.1. What portion(s) of the visual world does each functional cell type look at?

2.2. How consistent is this between animals?

2.3. What is the relationship in the portion(s) of the visual world of the different cell types sample?

2.4. To compare with retinal cell types, the receptive field overlap within and between different cell types should be presented.

3. The cluster analysis suggesting there are 24 functional cell types requires further support.

3.1. Please demonstrate that functional cell types are not highly over-represented in any single animal and that variability between animals is not a major contributor to the number of clusters.

3.2. It is not clear how separate the different clusters are. Could you please provide a visual representation of this? For example, scatter plots of the different clusters in feature space (like a pairs plot, e.g. seaborn pairplot or ggpairs in R), or demonstrate the separation using nonlinear embedding visualizations like tSNE.

4. There is currently very little evidence supporting the claim in lines 8-9 of the abstract "The second group is dominant at greater depths, consistent with a vertical progression of signal processing in the SC."

4.1. Further analysis should be provided to support this statement, or remove the claim.

5. The evidence supporting the reduction in dimensionality from the retina to the superior colliculus requires further support.

5.1. In panel 5D it is shown that there is a small decrease in the number of principal components needed to explain a certain level of variance. This is an interesting finding. Please demonstrate that this is not explained by the fact that the superior colliculus receives only a subset of visual information (~85% of the ganglion cells). Please report/estimate the dimensionality of the retinal ganglion cell types that project to the superior colliculus, rather than all retinal ganglion cells.

6. In panels 5B and 5C there is a presentation of the apparent opposite relationship between orientation and direction selectivity of neurons in the retina with a negative correlation and the superior colliculus with a positive correlation. This is a very interesting result. Could you please add the following to better support this assertion?

6.1. In Figure 5B please add an indication of which cell type is which, so it is easier to cross-reference with Baden et al. (2016). Do this also for Figure 5C so one can cross-reference with other figures.

6.2. Please also report this relationship for median OSI vs DSI in Figure 5B and 5C. In Baden et al. 2016 and figure 1 many of the distributions for OSI and DSI are highly skewed. It is quite amazing that no cell type has a low OSI and low DSI.

6.3. Why only report the relationship for 32 of the RGCs? They report 39 clusters of "certain" RGCs (Figure 2), where different clusters of the same group have different OSi vs DSi relationships.

6.4. In figure 11 please report E – G as 2D histograms/density plots. There are too many points to visualize the relationship.

6.5. Please re-plot the data in Figure 11 to demonstrate the relationship of DSI vs OSI of individual neurons by cluster. Is the relationship strongly positive for each cluster?

6.6. In the discussion, please provide references to other sources that the relationship between OSI and DSI in the retinal ganglion cells does indeed have a negative correlation.

*Reviewer #3 (Recommendations for the authors):*

Clustering approach: The steps between the measured calcium responses and the functional clusters in Figure 2 need to be described in more detail. Figure 6 is critical in this regard, but at present, it is quite hard to understand. One suggestion is to start with a measured calcium trace, show it in relation to several of the temporal features identified, and then show the decomposition of the response into features as in Figure 6. The leap currently straight to the feature representation in Figure 6A is too much. The figure caption could correspondingly be improved. Understanding this process is critical to evaluating the paper.

Number of clusters: Figure 6 shows that considering responses to individual stimuli results in more clusters than considering responses to the collection of stimuli together. This is hard to understand. Why doesn't the clustering using the full collection of responses project those responses down to the features identified, for example, for the chirp alone? From Figure 6 that would appear to increase the separable clusters. Related, why doesn't the ability to identify ~30 clusters for some stimuli indicate that there are ~30 cell types in SC? This result is not discussed sufficiently, and it would seem to bring into question one of the main results in the paper. Generally, the dependence of the number of clusters on the number and type of stimuli needs to be considered.

Figures: some of the figures are too small to appreciate. Some examples follow. Figure 1: the text above the traces in D is too small, and the directional selectivity and receptive field panels are too small. Figure 2: panel A is too packed and individual panels are too small. Figure 7: summary indices could be removed since they are already in Figure 2; doing so would create more space for the other panels. Axis labels are too small to easily read in several figures. It would also help to provide more information in several of the figures – e.g. in Figure 2F repeating the definition of the functional properties on the x-axis would be helpful.

Introduction: Suggest adding a sentence or two about the role of the SC in behavior.

Line 54: Suggest defining the stimuli in more detail.

Figure 2E: Can you flip the y-axis so that it corresponds (top to bottom) to Figure 2A?

Lines 108-109: Can you point out cells that have both orientation and directional tuning?

Section 2.3: This section discusses both visual space and space in the circuit (retina or SC). It is at times unclear which is being described. Related to this point, the Discussion makes a point about clustering in visual space, which should be made more clearly in this section of Results.

Section 2.4: Were the functional properties of cells with a specific genetic label similar (aside from RF sizes)?

Section 2.4 ends on a rather detailed point. Suggest coming back to differences between excitatory and inhibitory cells (or some other higher level point).

Lines 201-202: Can you add the identity of the RGC types corresponding to the clusters mentioned here?

[Editors' note: further revisions were suggested prior to acceptance, as described below.]

Thank you for resubmitting your work entitled "Functional Cell Types in the Mouse Superior Colliculus" for further consideration by *eLife*. Your revised article has been evaluated by Claude Desplan (Senior Editor) and a Reviewing Editor.

Most of the issues raised in the first round of review have been resolved. A few issues remain:

Line 30: Clarify whether the 10 types described here are in the superficial layer or entire SC.

Line 88: How are OSI and DSi incorporated into the distance measure? Are they weighted the same as the other parameters, or reweighted so that they have a larger impact on the distance measure?

Line 107: Can you state the difference between Figure 3B and 10A briefly in the main text so it is clearer why both are shown?

Line 137-138: Suggest stating the expected shape of the autocorrelation if the cells are randomly distributed. This would help interpret the decay in autocorrelation described here.

Line 160: What is "significant spatial separation" – is this a statistical statement?

Figure 6A: suggest flipping y-axis so that it corresponds to type 1 and type 2 groups in earlier figure. Text labels specifying SC type 1 and type 2 would also help.

The legends of the supplementary figures could get expanded considerably. At present they are quite brief and some of those figures are difficult to appreciate.

The format of the revised paper was inconvenient. The figure legends were embedded in the text, but the figures came at the end. Most importantly, the figures were not in the same order as the figure legends and were not themselves labeled (this had to do with where the supplementary figures ended up). Please provide an easier to read version when you resubmit!

---

## [Author Response]

Essential revisions:The reviewers all agreed that the data presented in the paper are quite valuable, but that several issues need to be improved before the paper can be fully evaluated. Chief among these are:1. Results from the genetic lines are not integrated well with the rest of the paper, and this misses an opportunity to provide more information about the properties of the labeled cells and how they fit into the remainder of the recorded population.

We have addressed this issue in responding to specific concerns raised by the reviewers.

2. Some key statements lack confirming analyses – e.g. the separation of cells into those that respond to the chirp stimulus and those that do not.

We have added more statistical analysis as requested by the reviewers.

3. Related to the point above, the paper contains several general statements about how SC is organized (e.g. the number of cell types and spatial organization of these types). The paper would benefit from focusing on those that are directly supported by the results and appropriate analyses. It is not entirely clear which those are without seeing how additional analyses turn out.

We have performed more analyses and focused the claims accordingly.

These and other important points are detailed in the individual reviews below.Reviewer #1 (Recommendations for the authors):An example is l. 93-99, where the authors say that "Group 1 … is distinct from Group 2 … in that it responds more strongly to the chirp stimuli", but this statement is not accompanied by any quantification.

We have compared the mean responses to the chirp stimuli between the two groups and found Group 1 responds significantly stronger than Group 2 (0.14 ± 0.05 vs 0.04 ± 0.04, p<0.001, t test).

"Almost all these types are excited by both On and Off stimuli” – do the authors refer to the responses to the steps in the chirp or any other part of the data?

Yes, we refer to the responses to the steps in the chirp; this is now clarified in the text.

No quantification/statistics are given for the statements regarding the differential sensitivity to temporal frequency.

The preference for temporal frequency was quantified by the frequency selectivity index. Group 1a is more selective to low frequency than Group 1b (p=0.02, Wilcoxon rank-sum test).

Similar in l. 107 ff, where statements about strong correlations are made, but not quantified.

The correlation was quantified by the correlation coefficients, and only significant values were color-coded in Figure 2F (new Figure 3C), with clarifications in the text and figure captions.

Or in l. 203, where it is stated that the sign of contribution changes between groups 1 and 2. Please add such information here and throughout.

We have revised this text and removed the claims that were not supported by the quantifications.

Several parts of the manuscript are directly related to the recently published findings from the Kremkow lab (10.1038/s41467-022-32775-2). While I assume that the questions have evolved independently in the two labs and that the data have been collected in parallel, I still find it somewhat disappointing that the important work from a junior PI is not cited in the manuscript. Beyond acknowledging the achievement, it would be worth discussing the similarities and differences between the two studies, which I perceive as clearly complementary. Similarly, I think the Baden dataset deserves a citation also in text around l. 196 (not only in the figure legend), and the idea of fitting higher visual responses for a weighted sum of RGC inputs was previously performed for dLGN in Roman-Roson et al., 2019.

We note that the Kremkow paper appeared only after we submitted our manuscript. These are all useful references, and they are cited in the revised text, including a comparison between our study and work from the Kremkow lab.

The authors use an elegant approach to imaging SC while leaving the cortex intact, but this comes with the caveat that only a small portion of SC is accessible. The authors touch upon this point but should discuss in more depth to which degree some of the features they observe might or might not be related to this caveat.

We have provided more discussion on this issue.

I think that an asset of the paper is the imaging of cre mouse lines, but I also have a number of questions. While the authors stated that in the overall population, they have mostly responses to on and off, they find off-type responses in some of the cre lines (e.g., l. 173). Why is this the case?

We have removed that statement, following further analysis of contrast selectivity for individual neurons.

Can the authors rule out that neurons with on or off responses are mixed in the clusters presented in Figure 2, and thus the cluster response is on-off?

We calculated the contrast selectivity index for each individual neuron, and conclude that the on-off character of a cluster’s average response is not the result of mixing On and Off cells. Please see the single-mode histograms in the CSI column of Figure 3A.

Also, instead of saying that the number of neurons is too low for clustering the data from the cre lines, could the authors try to find the best-matching cluster of the 24 clusters obtained in the large population of neurons?

We have added a new figure on this point (new Figure 5D).

Reviewer #2 (Recommendations for the authors):Li and Meister recorded the visual responses of more than 5000 neurons from the visual layers of the superior colliculus of 16 awake head-fixed mice allowed to run on a disc. Clustering of the visual responses suggests a greater diversity of functional cell types (24) in the superior colliculus than previously suggested (5 – 10). Neurons fell into two groups, those that respond similarly to retinal ganglion cells and a second group that showed more specialized stimulus selectivity. By comparing this data set with comparable recordings from the mouse retina (Baden et al., 2016) the authors suggest two differences: (i) The visual representation within the superior colliculus has a lower dimensionality than in the retina, (ii) In contrast to the retina, neurons of the same functional type form local clusters. This dataset has the potential to add important details to our understanding of how visual response properties are organized within the superior colliculus and highlight some potentially important key differences with the retina.While this paper has collected impressive data, there are two significant weaknesses that limit the impact of this work:1. Clear information about the relationship between the functional classes and genetically labeled populations.

We have provided such information in the new Figure 5D.

2. Clear description of how the different functional cell types are distributed across visual space.

Please see our responses below.

Providing a link to genetically targetable neuronal components of the superior colliculus would dramatically increase the usability of the data while describing how the set of functional cell types is distributed across the visual field would provide biological insight into the functional organization of the mouse superior colliculus.1. The integration of the data from the different genetically targeted cell types into the complete data set is not explicit or transparent. As currently presented it is unclear if this is exclusively an issue of analysis or a limitation of the data. An understanding of how functional classes are represented within and among different genetic cell types would make the dataset much more useful to the community and help to better assess the utility of these mouse lines. We have the following questions and requests.1.1. Please provide information about the distribution of the 24 functional classes for each genetic cell type.

The new Figure 5D shows such information for Cre-lines in which we collected data from at least three animals.

1.2. How separate are the different genetic cell classes?1.3. Is the functional segregation/overlap consistent with the anatomical overlap of these cell types (e.g Ntsr, tac1, and VGlut2, Zhou et al., 2017; Vgat and Rorb, Gale and Murphy, 2018)?1.4. What is the degree of anatomic overlap between genetic lines?1.5. Does this predict the amount of functional overlap?1.6. Please provide the anatomic distribution of the different genetic lines within the superficial layers of the colliculus.

VGluT2^+^ neurons are excitatory and Vgat+ neurons are inhibitory, so they are exclusive to each other. *Rorb*+ labels both excitatory and inhibitory neurons (Gale and Murphy, 2018). Ntsr1+ neurons are excitatory and show wide-field morphology (Gale and Murphy, 2018). Ntsr1+ neurons locate at the deeper part of superficial layer, and Vgat+ neurons are found across all the depth (Gale and Murphy, 2014) with the number decreasing slightly (Mize, 1992).

The new Figure 5D shows that more Vgat+ neurons belong to Group 1 that dominate the upper superficial layer. We provide this background information in the text (Sec 2.4). As regards the relation between anatomical and functional overlap, we don’t really have a hypothesis for such a relationship.

1.7. Please report the number of cells and animals recorded for each cell type in Figure 4.

We have added these numbers to the Methods.

2. A second issue in the paper is that the distribution of the different functional cell types in visual space is not analyzed. While it is reported in Figure 10 this representation does not report a systematic or unified analysis. In Figure 10 it is suggested that the recording technique measures from ~ 50 – 140 degrees of visual space horizontally and -10 to +60 degrees of visual space vertically. This appears to offer the opportunity to report useful information.2.1. What portion(s) of the visual world does each functional cell type look at?2.2. How consistent is this between animals?2.3. What is the relationship in the portion(s) of the visual world of the different cell types sample?

We offer data related to these points in the new Supplementary Figure 12. Also, we re-assessed the data set with these questions in mind and concluded that it is insufficient to sustain any strong claims about the visual fields of different functional types. The issue is that in each mouse the imaging window is in a slightly different place, and thus a different section of the visual field is covered. The regional variation in expression of the indicator contributes further variability across animals. This makes it difficult to piece together a global layout of functional types in the visual field from many local windows from different animals. We agree that the distribution of different processing functions across the visual field is an interesting subject, but it would benefit from a more global recording method. By contrast, we can be confident about the anatomical relationship of functional types within the same imaging window, and our report focuses on those.

2.4. To compare with retinal cell types, the receptive field overlap within and between different cell types should be presented.

We have added the RF overlap between different cell types to Figure 4D.

3. The cluster analysis suggesting there are 24 functional cell types requires further support.3.1. Please demonstrate that functional cell types are not highly over-represented in any single animal and that variability between animals is not a major contributor to the number of clusters.

We now report how each functional type derives from the various animals (Figure 12). Indeed, a few clusters are dominated by one or two animals. We have flagged those clusters in Figure 2, along with others that fail the Jacard criterion of robustness. One animal contributed about 25% of the recordings, owing to efficient expression of the indicator over a wide area of the SC. If we eliminate that animal from consideration, the number of functional types drops from 24 to 19, using the same BIC criterion. We mention in the corresponding Results section that the number of functional types is not constrained to high precision, but the data indicate a number near 20.

3.2. It is not clear how separate the different clusters are. Could you please provide a visual representation of this? For example, scatter plots of the different clusters in feature space (like a pairs plot, e.g. seaborn pairplot or ggpairs in R), or demonstrate the separation using nonlinear embedding visualizations like tSNE.

We have added a figure showing the relative positions of each cluster in the first 2 principal axes of feature space (new Figure 2C). This is similar to the cluster visualization for RGCs in Baden et al. 2016.

4. There is currently very little evidence supporting the claim in lines 8-9 of the abstract "The second group is dominant at greater depths, consistent with a vertical progression of signal processing in the SC."4.1. Further analysis should be provided to support this statement, or remove the claim.

We subjected this to further analysis. 68% of deeper neurons are from Group 2 (p<0.001, chisquare test).

5. The evidence supporting the reduction in dimensionality from the retina to the superior colliculus requires further support.5.1. In panel 5D it is shown that there is a small decrease in the number of principal components needed to explain a certain level of variance. This is an interesting finding. Please demonstrate that this is not explained by the fact that the superior colliculus receives only a subset of visual information (~85% of the ganglion cells). Please report/estimate the dimensionality of the retinal ganglion cell types that project to the superior colliculus, rather than all retinal ganglion cells.

Yes, it is thought that ~10% of RGCs avoid the superior colliculus. It is not entirely clear which RGC types those are. In at least one case (SPIG1+ cells) there appears to be a regional difference across the retina within the same RGC type. So certainly part of the reduction in dimensionality could result from selection of the afferents, but not enough is known about the cell types involved for a precise calculation. We have added this caveat to the discussion.

6. In panels 5B and 5C there is a presentation of the apparent opposite relationship between orientation and direction selectivity of neurons in the retina with a negative correlation and the superior colliculus with a positive correlation. This is a very interesting result. Could you please add the following to better support this assertion?6.1. In Figure 5B please add an indication of which cell type is which, so it is easier to cross-reference with Baden et al. (2016). Do this also for Figure 5C so one can cross-reference with other figures.

We have added a color coding for this to the new Figure 6B and 6C.

6.2. Please also report this relationship for median OSI vs DSI in Figure 5B and 5C. In Baden et al. 2016 and figure 1 many of the distributions for OSI and DSI are highly skewed. It is quite amazing that no cell type has a low OSI and low DSI.

We have added that to the new Figure 6B and 6C.

6.3. Why only report the relationship for 32 of the RGCs? They report 39 clusters of "certain" RGCs (Figure 2), where different clusters of the same group have different OSi vs DSi relationships.

Baden 2016 reported 49 clusters, which were then combined into 32 types based on staining and morphology for better interpretation. Therefore we adopted 32 types rather than 49 clusters.

6.4. In figure 11 please report E – G as 2D histograms/density plots. There are too many points to visualize the relationship.

We have replotted the panels as suggested in the new Supplementary Figure 13.

6.5. Please re-plot the data in Figure 11 to demonstrate the relationship of DSI vs OSI of individual neurons by cluster. Is the relationship strongly positive for each cluster?

In the new Figure 13J, we show the correlation coefficient for each cluster.

6.6. In the discussion, please provide references to other sources that the relationship between OSI and DSI in the retinal ganglion cells does indeed have a negative correlation.

This result is based on our own analysis of Baden’s dataset. We are not aware of another report.

Reviewer #3 (Recommendations for the authors):Clustering approach: The steps between the measured calcium responses and the functional clusters in Figure 2 need to be described in more detail. Figure 6 is critical in this regard, but at present, it is quite hard to understand. One suggestion is to start with a measured calcium trace, show it in relation to several of the temporal features identified, and then show the decomposition of the response into features as in Figure 6. The leap currently straight to the feature representation in Figure 6A is too much. The figure caption could correspondingly be improved. Understanding this process is critical to evaluating the paper.

The calcium traces of individual neurons were and still are shown in Figure 1 and the new Supplementary Figure 9. We have expanded the text surrounding the feature matrix, in both methods and the caption of Figure 7. The dimensionality-reduction method we used is fairly standard.

Number of clusters: Figure 6 shows that considering responses to individual stimuli results in more clusters than considering responses to the collection of stimuli together. This is hard to understand. Why doesn't the clustering using the full collection of responses project those responses down to the features identified, for example, for the chirp alone? From Figure 6 that would appear to increase the separable clusters. Related, why doesn't the ability to identify ~30 clusters for some stimuli indicate that there are ~30 cell types in SC? This result is not discussed sufficiently, and it would seem to bring into question one of the main results in the paper. Generally, the dependence of the number of clusters on the number and type of stimuli needs to be considered.

These are good questions. The choice of stimuli will certainly influence the identification of functional types. Rather than devising a new stimulus set arbitrarily, we used the stimuli that had previously served to separate retinal ganglion cell types. This is relevant for two reasons: (1) In case of the retina there is ground truth about the identity of cell types, based on gene expression data and the anatomical tiling criterion. (2) RGCs are immediately presynaptic to the superior colliculus, and the transformation that occurs from retina to SC is of special interest here. Ultimately the number of clusters is also influenced by the Bayesian Information Criterion, that balances the goodness of the fit with the complexity of the model. A section in the Discussion brings up these caveats regarding the precise number of types from the cluster analysis.

Figures: some of the figures are too small to appreciate. Some examples follow. Figure 1: the text above the traces in D is too small, and the directional selectivity and receptive field panels are too small. Figure 2: panel A is too packed and individual panels are too small. Figure 7: summary indices could be removed since they are already in Figure 2; doing so would create more space for the other panels. Axis labels are too small to easily read in several figures. It would also help to provide more information in several of the figures – e.g. in Figure 2F repeating the definition of the functional properties on the x-axis would be helpful.

For Figure 1, we have deleted the text. For Figure 2, we have divided it into two figures. For Figure 7, we have removed the summary indices. We have also increased the size of labels as suggested.

Introduction: Suggest adding a sentence or two about the role of the SC in behavior.

We have added a sentence at the end of the first paragraph.

Line 54: Suggest defining the stimuli in more detail.

We added a brief description of stimuli to the figure caption. The detailed definition is in the Methods. We have indicated this in the text.

Figure 2E: Can you flip the y-axis so that it corresponds (top to bottom) to Figure 2A?

We have flipped the y-axis.

Lines 108-109: Can you point out cells that have both orientation and directional tuning?

Cells in types 9, 14, 20, 24 show both orientation and direction tuning. We have added that to the text.

Section 2.3: This section discusses both visual space and space in the circuit (retina or SC). It is at times unclear which is being described. Related to this point, the Discussion makes a point about clustering in visual space, which should be made more clearly in this section of Results.

This Results section is only about anatomical space in the SC. We revised the section title to make this clear.

Section 2.4: Were the functional properties of cells with a specific genetic label similar (aside from RF sizes)?

This information was provided in Figure 4B (new Figure 5B) with histograms for each functional property and genetic label. Some of these distributions are rather sharp, for example most *Rorb*+ neurons show very low orientation and direction selectivity. Others vary more broadly, for example the best stimulus size (BSS) covers quite a range within all the genetic types.

Section 2.4 ends on a rather detailed point. Suggest coming back to differences between excitatory and inhibitory cells (or some other higher level point).

We changed the ending as suggested.

Lines 201-202: Can you add the identity of the RGC types corresponding to the clusters mentioned here?

We have added their identifications.

[Editors' note: further revisions were suggested prior to acceptance, as described below.]

Most of the issues raised in the first round of review have been resolved. A few issues remain:Line 30: Clarify whether the 10 types described here are in the superficial layer or entire SC.

We have clarified that they are in the superficial SC.

Line 88: How are OSI and DSi incorporated into the distance measure? Are they weighted the same as the other parameters, or reweighted so that they have a larger impact on the distance measure?

They are weighted the same as the other parameters. All features were normalized in the same way, which is described in line 503 of Methods section 4.5.3.

Line 107: Can you state the difference between Figure 3B and 10A briefly in the main text so it is clearer why both are shown?

Figure 3B only shows values that are significant from zero while Figure 10A shows all values. We have revised the text accordingly.

Line 137-138: Suggest stating the expected shape of the autocorrelation if the cells are randomly distributed. This would help interpret the decay in autocorrelation described here.

We have revised the text accordingly.

Line 160: What is "significant spatial separation" – is this a statistical statement?

Yes. We have added the statistical information to the text and figure legend.

Figure 6A: suggest flipping y-axis so that it corresponds to type 1 and type 2 groups in earlier figure. Text labels specifying SC type 1 and type 2 would also help.The legends of the supplementary figures could get expanded considerably. At present they are quite brief and some of those figures are difficult to appreciate.

We have expanded some of these legends. Note also that each supplement figure is clearly associated with a main text figure.

The format of the revised paper was inconvenient. The figure legends were embedded in the text, but the figures came at the end. Most importantly, the figures were not in the same order as the figure legends and were not themselves labeled (this had to do with where the supplementary figures ended up). Please provide an easier to read version when you resubmit!

This format must have been constructed by the submission system by piecing together figures and text. We had also provided a properly formatted PDF file. We did the same on this revision.